# Gauge-Equivariant Graph Networks via Self-Interference Cancellation

Yoonhyuk Choi [1]   Jiho Choi [2]   Jiwoo Kang [1]

## Abstract

Graph neural networks often degrade on heterophilous graphs because repeated neighbor aggregation can reinforce self-aligned low-frequency components while suppressing phase-inconsistent signals. We propose GESC, a complex-valued graph network that augments attention-based message passing with gauge-consistent U(1) transport and projection-based self-interference cancellation. For each transported neighbor message, GESC removes the component parallel to the target representation before computing attention and applies a sign-aware gate based on gauge-invariant complex alignment. We prove gauge equivariance of the hidden update and derive coefficient-frozen stability bounds showing that SIC contracts self-parallel message components. On nine benchmarks, GESC ranks first on seven datasets and remains within the top three on the other two. These results suggest that explicit self-parallel cancellation is an effective mechanism for improving message passing under heterophily. Our code is available at this link.

## 1. Introduction

Graph Neural Networks (GNNs) have emerged as a powerful paradigm for learning on relational data, achieving state-of-the-art performance across social networks, recommendation, and scientific computing (Kipf & Welling, 2017; Veličković et al., 2018; Gilmer et al., 2017). Despite these successes, most GNNs rely on real-valued message passing that implicitly assumes homophily, symmetric interactions, and non-negative aggregation (Wu et al., 2019b; Bronstein et al., 2021). This inductive bias aligns well with classical benchmarks but fails to represent the directional or oscillatory nature of many real-world systems.

[1]Division of Artificial Intelligence, Sookmyung Women's University, Seoul, South Korea [2]Korea Advanced Institute of Science and Technology, Seoul, South Korea. Correspondence to: Jiwoo Kang <jwkang@sookmyung.ac.kr>.

*Proceedings of the 43rd International Conference on Machine Learning*, Seoul, South Korea. PMLR 306, 2026. Copyright 2026 by the author(s).

Many graphs exhibit interfering interactions that cannot be faithfully captured by additive aggregation. Such phenomena arise in systems governed by transport, diffusion, or wave propagation, and can be modeled through magnetic Laplacians, signed-directed operators, or high-frequency spectral components (Levie et al., 2018; Rusch & Mishra, 2020; Marques et al., 2020; Wang et al., 2021; Lim et al., 2022; Ko et al., 2023). However, standard message passing aggregates neighbor signals indiscriminately, amplifying redundant components and suppressing antisymmetric structure. Under this condition, repeated message-passing reinforces self-parallel information over multiple hops, which can accumulate biases (over-smoothing) (Li et al., 2018; Oono & Suzuki, 2019; Alon & Yahav, 2021; Black et al., 2023) and degrade under heterophily (Zhu et al., 2020; Lim et al., 2021). This failure to represent destructive interference limits GNN expressivity in non-homophilous settings.

Recent studies have sought to alleviate these issues through more expressive aggregation using signed messaging (Huang et al., 2019; Li et al., 2022b; Choi et al., 2025a), heterophily-aware architectures (Jin et al., 2021; Choi et al., 2025b), adaptive or spectral filtering (Chien et al., 2021; He et al., 2021; Ma et al., 2021), or decomposition-based designs (Wang & Derr, 2021; Yan et al., 2022). While these models mitigate over-smoothing, they remain fundamentally scalar reweighting schemes that fail to represent the phase structure of interactions. Without explicit modeling of relative phases or destructive alignment, such approaches cannot express oscillatory or signed behaviors essential for directional and anti-homophilous graphs (Levie et al., 2018; Xu et al., 2019; de Haan et al., 2021; Brandstetter et al., 2022; Zheng et al., 2022; Rampášek et al., 2022). This motivates a principled mechanism that directly models and cancels interference, rather than merely reweighting it away.

To realize this idea, our model represents nodes as complex embeddings and equips each edge with a learnable $U(1)$ phase connection with gauge-equivariant transport (de Haan et al., 2021; Brandstetter et al., 2022). Unlike prior magnetic Laplacian-based or gauge-equivariant GNNs (Zhang et al., 2021; He et al., 2022), which primarily achieve phase-consistent message passing through scalar reweighting, we introduce a projection-based self-interference cancellation that enables true gauge- and permutation-equivariant filtering within the complex message space. Our interference-

aware design removes redundant components at their source and aligns neighbor messages through magnetic transport and sign-aware gating. Specifically, (i) a projection-based Self-Interference Cancellation (SIC) removes redundant components before attention; (ii) a sign-aware gate modulates neighbors based on their relative phase; and (iii) an interference-aware message passing achieves stable propagation and mitigates over-smoothing at finite depth. Our contributions are threefold:

- We introduce Self-Interference Cancellation (SIC), a projection-based mechanism that removes redundant self-aligned components, mitigating constructive accumulation bias and reducing over-smoothing in GNNs.

- We propose a Sign-aware Gating with Gauge Transport mechanism that leverages complex phase alignment for smooth and stable modulation without relying on explicit direction labels.

- We provide theoretical analysis and extensive empirical evaluation on synthetic and real-world graph benchmarks, demonstrating improved stability, heterophily robustness, and state-of-the-art performance.

## 2. Related Work

**Graph Neural Networks and Heterophily.** Classical GNNs are effective in homophilous regimes, but degrade on heterophilous graphs (Defferrard et al., 2016; Kipf & Welling, 2017; Veličković et al., 2018; Gilmer et al., 2017; Abu-El-Haija et al., 2019). To address this, a large body of work introduces signed or adaptive propagation mechanisms (Derr et al., 2018; Klicpera et al., 2019; Gasteiger et al., 2019; Huang et al., 2019; Bo et al., 2021; Chien et al., 2021; Du et al., 2022; Choi et al., 2025a). Recent studies also propose link-level features (Lim et al., 2021) and decomposition-based heterophily models (Wang & Derr, 2021; Yan et al., 2022). Unlike these approaches, our formulation targets heterophily through geometric interference control rather than edge reweighting, explicitly cancelling redundant self-components via rank-1 projection before attention and applying sign-aware gating.

**Complex and Spectral-Phase Representations.** A second line of work leverages complex embeddings to encode antisymmetry (Trouillon et al., 2016; Sun et al., 2019). Early spectral (Levie et al., 2018) and more recent graph networks (Zhang et al., 2019; 2022) demonstrate that complex-valued filtering can naturally encode phase and orientation. Magnetic Laplacian formulations introduce complex Hermitian operators (Zhang et al., 2021; He et al., 2022), and recent spectral designs extend this idea to richer frequency responses (Rusch & Mishra, 2020). Most prior works incorporate phase information only at the filtering level or as an auxiliary feature. In contrast, we inject phase alignment into spatial attention and guarantee gauge-invariant scoring.

**Gauge Equivariance and Phase Transport.** Building on these phase-based designs, gauge-equivariant networks formalize invariance to local reference frames (Cohen & Welling, 2016; Cohen et al., 2019; de Haan et al., 2021), with recent developments connecting geometric deep learning to physical principles (Pei et al., 2020; Bronstein et al., 2021; Brandstetter et al., 2022; Luo et al., 2022). We reinterpret edge phases as latent directional transport to ensure scalar scores and alignments remain gauge-invariant within our gating mechanism. This treatment complements magnetic and spectral designs (Marques et al., 2020; Zhang et al., 2021; He et al., 2022; Lim et al., 2022), but differs by learning phase transport end-to-end in the spatial domain.

**Over-smoothing and Interference.** Deep message passing often causes representation homogenization (Li et al., 2018; Oono & Suzuki, 2019; Alon & Yahav, 2021; Black et al., 2023), limiting expressivity. Typical fixes include residual connections, normalization, or hop-wise feature mixing (Klicpera et al., 2019; Gasteiger et al., 2019; Frasca et al., 2020; Wang et al., 2021). Our Self-Interference Cancellation (SIC) module takes a geometric approach: it projects out the transported neighbor component parallel to the target state before attention, mitigating self-reinforcement and preserving orthogonal signals. This stands in contrast to heuristic ego-neighbor separation methods such as H₂GCN (Zhu et al., 2020), offering a principled interference control mechanism tied to phase alignment.

More details can be found in Appendix A.

## 3. Preliminaries

**Graph structure.** Let $\mathcal{G} = (\mathcal{V}, \mathcal{E}, X)$ denote a finite undirected attributed graph to establish notation for subsequent sections. The node feature matrix $X \in \mathbb{R}^{N \times d_{\text{in}}}$ encodes $d_{\text{in}}$-dimensional real-valued features for each node. The topology is represented by a symmetric adjacency matrix $A \in \{0,1\}^{N \times N}$ with $A_{ij} = A_{ji} = 1$ if and only if $\{i,j\} \in \mathcal{E}$. We denote by $D$ the diagonal degree matrix $D_{ii} = \sum_{j=1}^{N} A_{ij}$ and by $\mathcal{N}(i)$ the neighbor set of node $i$. Each node $i$ has a label $y_i \in \{1, \ldots, C\}$, and $Y \in \mathbb{R}^{N \times C}$ is the one-hot label matrix. Following (Zhu et al., 2020), the global homophily level is given by:

$$\mathcal{G}_h := \frac{1}{|\mathcal{E}|} \sum_{\{i,j\} \in \mathcal{E}} \mathbb{I}(y_i = y_j), \qquad (1)$$

where lower $\mathcal{G}_h$ indicates stronger heterophily. We focus on semi-supervised node classification with a labeled subset $\mathcal{V}_L$ and unlabeled nodes $\mathcal{V}_U = \mathcal{V} \setminus \mathcal{V}_L$.

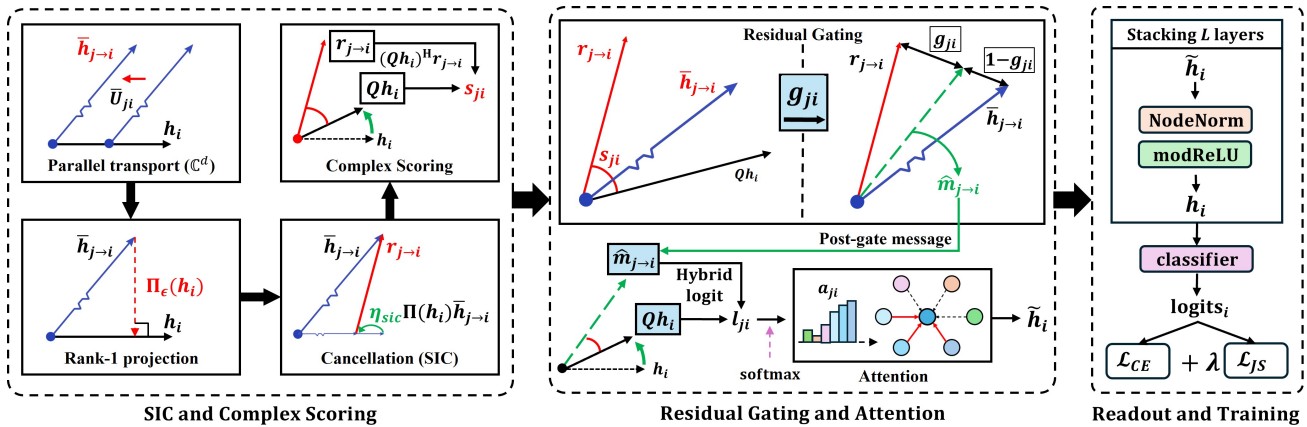

*Figure 1.* Overview of GESC. (*Left*) Neighbor messages are parallel transported and self-parallel components are cancelled via rank-1 projection, yielding gauge-invariant complex scores. (*Middle*) A sign-aware residual gate softly interpolates messages before hybrid magnitude-phase attention. (*Right*) Aggregated features undergo NodeNorm and modReLU, followed by classification with cross-entropy and JS consistency.

**Complex node embeddings and notation.** We operate in a complex vector space $\mathbb{C}^d$, which naturally encodes both magnitude and phase information. Each node $i$ carries an embedding $h_i^{(t)} \in \mathbb{C}^d$ at layer $t$, where the entire node representation is defined as follows:

$$H^{(t)} = \begin{bmatrix} (h_1^{(t)})^\top \\ \vdots \\ (h_N^{(t)})^\top \end{bmatrix} \in \mathbb{C}^{N \times d}. \qquad (2)$$

The initial embedding is obtained by linearly mapping real-valued features into the complex domain using two real weight matrices for the real and imaginary parts. We denote the conjugate transpose by $(\cdot)^{\mathrm{H}}$, the elementwise complex conjugate by $\overline{(\cdot)}$, the transported quantities by $\tilde{\cdot}$ (tilde), and $\hat{\cdot}$ (hat) for post-gating messages.

**Complex inner products and projections.** For the nodes $u, v \in \mathbb{C}^d$, the complex inner product is defined as

$$\langle u, v \rangle_{\mathbb{C}} = u^{\mathrm{H}} v = \sum_{k=1}^{d} \overline{u_k} v_k, \quad \|u\|_2 = \sqrt{\langle u, u \rangle_{\mathbb{C}}}. \quad (3)$$

We use a Tikhonov-regularized rank-1 projector $\Pi_\epsilon(h_i)$ onto the one-dimensional subspace spanned by $h_i$ to separate the self-parallel and orthogonal components of messages:

$$\Pi_\epsilon(h_i) = \frac{h_i h_i^{\mathrm{H}}}{\|h_i\|_2^2 + \epsilon}, \qquad (4)$$

with $\epsilon > 0$ for numerical stability. This operator will play a central role in the self-interference cancellation (SIC) mechanism introduced in Section 4.

**Edge transport and phase structure.** Each undirected edge $\{i, j\}$ is assigned a unit-magnitude complex phase

factor $\theta_{ji} \in \mathbb{R}$, which induces a directed pair of parallel transports with opposite phases:

$$U_{ji} := e^{i\theta_{ji}}, \quad U_{ij} = \overline{U_{ji}} = e^{-i\theta_{ji}}, \qquad (5)$$

where $|U_{ji}| = |U_{ij}| = 1$. We use complex weights $W, Q \in \mathbb{C}^{d \times d}$ (indices omitted here for clarity). Then, the transported source in $i$'s reference frame is given by:

$$\tilde{h}_{j \to i} = U_{ji} W h_j. \qquad (6)$$

**Gauge-invariant formulation.** For arbitrary phases $\{\psi_i\}$, we adopt a local U(1) gauge symmetry:

$$h_i \mapsto h_i' = e^{i\psi_i} h_i, \qquad (7)$$
$$(\theta_{ji}, U_{ji} = e^{i\theta_{ji}}) \mapsto (\theta_{ji} + \psi_i - \psi_j, \, e^{i(\psi_i - \psi_j)} U_{ji}).$$

Then, $\tilde{h}_{j \to i} \mapsto \tilde{h}'_{j \to i} = e^{i\psi_i} \tilde{h}_{j \to i}$ (gauge-equivariant). If $W, Q$ are gauge-invariant linear maps, scalar alignments like

$$\left(Q h_i\right)^{\mathrm{H}} \tilde{h}_{j \to i} \qquad (8)$$

are exactly gauge-invariant, because both factors acquire the same local phase $e^{i\psi_i}$ which cancels out in the Hermitian product. This invariance ensures that all subsequent attention logits and projection scores in Section 4 are computed within a locally gauge-consistent reference frame.

## 4. Methodology

### 4.1. Overview

We replace additive neighbor accumulation in GNNs with a wave-interference mechanism that (i) cancels redundant self-components before attention, (ii) softly modulates neighbor messages with a sign-aware gate to reduce harmful influence

without over-suppressing negative correlations, and (iii) leverages complex-valued representations with magnitude-robust nonlinearities and optional phase-aware attention. To further stabilize sign-aware gating under noisy neighbors, we employ a magnetic transport $U_{ji} = e^{i\theta_{ji}}$ in Eq. 5, which ensures gauge-invariant alignment and improves robustness in practice. All operations follow the gauge transformation convention, preserving gauge invariance of scalar scores and alignment terms.

### 4.2. Proposed Method

**SIC and Complex Scoring.** Each layer uses $M$ heads. Head $m$ applies a source transport $W^{(m)} \in \mathbb{C}^{d \times d}$ and a target filter $Q^{(m)} \in \mathbb{C}^{d \times d}$. For each neighbor $j \in \mathcal{N}(i)$ (induced direction $j \to i$), we use the unit-modulus transport defined in Eq. 5:

$$U_{ji} = e^{i\theta_{ji}}, \quad U_{ij} = \overline{U_{ji}}. \tag{9}$$

The transported source feature in $i$'s reference frame is

$$\tilde{h}_{j \to i}^{(m)} := U_{ji} W^{(m)} h_j^{(t)} \in \mathbb{C}^d. \tag{10}$$

To remove redundant self-parallel components before attention, we apply a Tikhonov-regularized rank-1 projector onto the target state $h_i^{(t)}$:

$$\Pi_\epsilon(h_i^{(t)}) := \frac{h_i^{(t)} (h_i^{(t)})^{\mathrm{H}}}{\|h_i^{(t)}\|_2^2 + \epsilon}, \tag{11}$$

$$r_{j \to i}^{(m)} := \tilde{h}_{j \to i}^{(m)} - \eta_{\mathrm{sic}} \cdot \Pi_\epsilon(h_i^{(t)}) \tilde{h}_{j \to i}^{(m)}, \tag{12}$$

where $\eta_{\mathrm{sic}} \in [0, 1]$ controls cancellation strength and $\epsilon > 0$ ensures numerical stability. The complex alignment score is

$$s_{ji}^{(m)} := (Q^{(m)} h_i^{(t)})^{\mathrm{H}} r_{j \to i}^{(m)} \in \mathbb{C}. \tag{13}$$

By the U(1) gauge convention, $s_{ji}^{(m)}$ is a gauge-invariant scalar. For brevity, we denote the magnitude product used for normalized real alignment in the gate as

$$\nu_{ji}^{(m)} := \|Q^{(m)} h_i^{(t)}\|_2 \|r_{j \to i}^{(m)}\|_2 + \epsilon. \tag{14}$$

**Sign-aware (phase-consistent) gating.** We modulate the SIC-residual $r_{j \to i}^{(m)}$ by a sign-aware scalar gate $\xi_{ji}^{(m)} \in [0, 1]$ computed from a gauge-invariant alignment:

$$\rho_{ji}^{(m)} := \mathrm{Re}\left(\frac{s_{ji}^{(m)}}{\nu_{ji}^{(m)}}\right), \tag{15}$$

$$\xi_{ji}^{(m)} := \sigma(c_m \rho_{ji}^{(m)} + d_m), \tag{16}$$

where $c_m, d_m$ are learnable scalars and $\sigma(x) = \frac{1}{1+e^{-x}}$. The sign-aware residual is given by:

$$\bar{r}_{j \to i}^{(m)} := \xi_{ji}^{(m)} r_{j \to i}^{(m)}. \tag{17}$$

**Residual Gating and Alignment-based Attention.** Each head learns a residual mixing gate $g_{ji}^{(m)} \in [0, 1]$:

$$g_{ji}^{(m)} = \sigma\left(a_m^\top [\varphi(\|\bar{r}_{j \to i}^{(m)}\|_2), \varphi(\|\tilde{h}_{j \to i}^{(m)}\|_2), \varphi(|s_{ji}^{(m)}|)]\right), \tag{18}$$

where $\varphi(x) = \log(1 + x)$. The post-gate message is

$$\widehat{m}_{j \to i}^{(m)} := g_{ji}^{(m)} \bar{r}_{j \to i}^{(m)} + (1 - g_{ji}^{(m)}) \tilde{h}_{j \to i}^{(m)}. \tag{19}$$

Attention logits are computed from the complex alignment between the target and post-gate message:

$$\tilde{s}_{ji}^{(m)} := (Q^{(m)} h_i^{(t)})^{\mathrm{H}} \widehat{m}_{j \to i}^{(m)}. \tag{20}$$

For the attention normalization, we use a separate magnitude product with a hybrid logit that combines magnitude and normalized real alignment:

$$\tilde{\nu}_{ji}^{(m)} := \|Q^{(m)} h_i^{(t)}\|_2 \|\widehat{m}_{j \to i}^{(m)}\|_2 + \epsilon. \tag{21}$$

$$\ell_{ji}^{(m)} = \gamma_m \left[\lambda \frac{|\tilde{s}_{ji}^{(m)}|}{\sqrt{d}} + (1 - \lambda)\mathrm{Re}\left(\frac{\tilde{s}_{ji}^{(m)}}{\tilde{\nu}_{ji}^{(m)}}\right)\right], \tag{22}$$

where $\gamma_m > 0$ is a learnable temperature and $\lambda \in [0, 1]$ balances magnitude and phase contributions. Because $\widehat{m}_{j \to i}^{(m)}$ is obtained from gauge-equivariant $r_{j \to i}^{(m)}$ via scalar gates, $\tilde{s}_{ji}^{(m)}$ remains gauge-invariant. The attention weights and aggregated message are

$$\alpha_{ji}^{(m)} = \mathrm{softmax}_{j \in \mathcal{N}(i)}(\ell_{ji}^{(m)}), \tag{23}$$

$$\widetilde{h}_i^{(t+1)} = h_i^{(t)} + \sum_{m=1}^{M} \sum_{j \in \mathcal{N}(i)} \alpha_{ji}^{(m)} \widehat{m}_{j \to i}^{(m)}. \tag{24}$$

**Readout and Training.** We apply node-wise normalization and complex nonlinearity:

$$h_i^{(t+1)} = \mathrm{modReLU}(\mathrm{NodeNorm}(\widetilde{h}_i^{(t+1)})). \tag{25}$$

Here, the $\mathrm{NodeNorm}(\cdot)$ and $\mathrm{modReLU}(\cdot)$ are given by:

$$\mathrm{NodeNorm}(h_i) = \frac{h_i - \mu_i}{\varsigma_i + \epsilon}, \tag{26}$$

$$\mathrm{modReLU}(z) = \max(|z| + b, 0) \cdot \frac{z}{|z| + \epsilon}, \tag{27}$$

where $b$ is a (real-valued) learnable bias, $\mu_i = \frac{1}{d} \sum_{k=1}^{d} h_{ik}$, and $\varsigma_i = \sqrt{\frac{1}{d} \sum_{k=1}^{d} |h_{ik} - \mu_i|^2}$. For classification, the final complex node representations $h_i^{(L)}$ are mapped to real logits by concatenating their real and imaginary parts, followed by LayerNorm and a linear classifier:

$$z_i = \mathrm{vec}(\mathrm{Re}(h_i^{(L)}), \mathrm{Im}(h_i^{(L)})) \in \mathbb{R}^{2d}, \tag{28}$$

$$\mathrm{logits}_i = W_{\mathrm{cls}} \mathrm{Dropout}(\mathrm{LayerNorm}(z_i)) \in \mathbb{R}^C.$$

We train the model with a cross-entropy loss on labeled nodes $\mathcal{V}_L$:

$$\mathcal{L}_{\text{CE}} = -\frac{1}{|\mathcal{V}_L|} \sum_{i \in \mathcal{V}_L} \log p_\Theta(y_i \mid \text{logits}_i), \qquad (29)$$

and a Jensen-Shannon consistency term between two stochastic forward passes with temperature $T > 0$:

$$\mathcal{L}_{\text{JS}} = \text{JS}\left(\text{softmax}(\text{logits}^{(1)}/T), \text{softmax}(\text{logits}^{(2)}/T)\right). \tag{30}$$

The final training objective is defined as below:

$$\mathcal{L} = \mathcal{L}_{\text{CE}} + \lambda_{\text{JS}} \mathcal{L}_{\text{JS}}. \tag{31}$$

The overall algorithm and implementation details are given in Appendix B.

### 4.3. Scalability on Large Graphs

For each head $m$, one GESC layer involves several computational steps. First, applying the complex linear transform $W^{(m)} \in \mathbb{C}^{d \times d}$ to all source embeddings costs $O(Ed^2)$ when performed directly on edge-stacked features. Next, for each edge $(j, i)$, the SIC projection requires computing the scalar $(h_i^* \tilde{h}_{j \to i})$ and removing its parallel component. This involves one complex inner product and one scalar-vector multiply, resulting in $O(d)$ per edge and a total of $O(Ed)$. Attention computation, including magnitude calculation and degree-wise softmax normalization, adds $O(E)$, while message formation and aggregation contribute another $O(Ed)$. Summing these components and multiplying by $M$ attention heads, the overall forward complexity is $O(MEd^2)$, where the $d^2$ term arises from the complex linear transformations. Since $E$ typically scales much larger than $N$ in real-world sparse graphs and the hidden dimension $d$ remains moderate, this term dominates the runtime.

## 5. Theoretical Properties

In this section, we present theoretical justifications for the core design choices introduced in Section 4. Specifically, we show that (i) Self-Interference Cancellation (SIC) attenuates self-parallel components and mitigates low-frequency dominance, (ii) sign-aware gating provides bounded and stable propagation, (iii) complex-valued transport maintains the gauge structure and invariance, and (iv) the resulting GESC layer admits explicit Lipschitz bounds, contributing to robustness against spectral collapse and over-smoothing.

### 5.1. Why SIC Works: a Linear-Algebraic View

Compared to the prior works (Zhang et al., 2021; He et al., 2022), SIC addresses the accumulation bias, acting as a notch on low-frequency modes in self-aligned directions.

**Proposition 5.1** (Effect of SIC on message decomposition).
*Let $\tilde{h} = \tilde{h}_\parallel + \tilde{h}_\perp$ with $\tilde{h}_\parallel \parallel h_i^{(t)}$ and $\tilde{h}_\perp \perp h_i^{(t)}$. Using the Tikhonov-regularized rank-1 operator $\Pi_\epsilon(h_i^{(t)})$ in Eq. 11,*

$$(I - \eta_{\text{sic}}\Pi_\epsilon(h_i^{(t)}))\tilde{h} = \left(1 - \eta_{\text{sic}}\frac{\|h_i^{(t)}\|_2^2}{\|h_i^{(t)}\|_2^2 + \epsilon}\right)\tilde{h}_\parallel + \tilde{h}_\perp \tag{32}$$

*Thus, SIC attenuates the self-parallel component while preserving orthogonal ones. Proof is in Appendix C.1.1.*

**Lemma 5.2** (Reduction of self-parallel energy). *Let $\Pi_\epsilon(h_i^{(t)})$ be the Tikhonov-regularized rank-1 projector in Eq. 11, and define the self-parallel energy as follows:*

$$\mathcal{E}_\parallel(x) := \left\|\Pi_\epsilon(h_i^{(t)})x\right\|_2^2. \tag{33}$$

*For any $\eta_{\text{sic}} \in [0, 1]$ and any $\tilde{h} \in \mathbb{C}^d$,*

$$\mathcal{E}_\parallel\left((I - \eta_{\text{sic}}\Pi_\epsilon(h_i^{(t)}))\tilde{h}\right) \leq \mathcal{E}_\parallel(\tilde{h}) \tag{34}$$

*In particular, SIC does not increase the squared magnitude of the component parallel to $h_i^{(t)}$, since $\Pi_\epsilon(h_i^{(t)})$ is a positive semidefinite rank-1 projector with operator norm at most 1. This limits the influence of self-reinforcing directions, redistributing attention mass toward more informative neighbors. Proof is in Appendix C.1.2.*

**Remark 5.3** (Spectral notch effect). Consider a linearized propagation step with normalized adjacency $\tilde{A}$ and self-loop weight $\alpha$:

$$h^{(t+1)} \approx \alpha h^{(t)} + \tilde{A}h^{(t)}. \tag{35}$$

Under diffusion-like propagation, the dominant local direction of $h^{(t)}$ tends to align with low-frequency eigenmodes of the graph Laplacian. Applying the self-interference cancellation operator $(I - \eta_{\text{sic}}\Pi_\epsilon(h_i^{(t)}))$ before attention attenuates the component of each local message parallel to $h_i^{(t)}$, reducing the energy of these low-frequency modes. Consequently, SIC acts as a node-wise notch filter that suppresses low-frequency dominance and delays spectral collapse as depth increases. A more detailed discussion is provided in Appendix C.1.3.

### 5.2. Stability from Sign-aware Gating

Sign-aware gating provides an additional mechanism for stabilizing propagation by down-weighting negatively aligned residuals before attention. Below, we prove that the resulting gated messages remain uniformly bounded in norm.

**Proposition 5.4** (Per-head boundedness with sign-aware gating). *Assume $U_{ji}$ is unitary, $\eta_{\text{sic}} \in [0, 1]$, and $\xi_{ji}^{(m)}, g_{ji}^{(m)} \in [0, 1]$. Let $\{\alpha_{ji}^{(m)}\}_{j \in \mathcal{N}(i)}$ be the attention weights with $\alpha_{ji}^{(m)} \geq 0$ and $\sum_{j \in \mathcal{N}(i)} \alpha_{ji}^{(m)} = 1$. Then, for any node $i$ and head $m$,*

$$\left\| \sum_{j \in \mathcal{N}(i)} \alpha_{ji}^{(m)} \widehat{m}_{j \to i}^{(m)} \right\|_2 \leq \|W^{(m)}\|_2 \cdot \max_{j \in \mathcal{N}(i)} \|h_j^{(t)}\|_2. \tag{36}$$

*Proof is given in Appendix C.2.1.*

This per-head bound shows that the gated aggregation cannot grow faster than a linear map with operator norm $\|W^{(m)}\|_2$, ensuring stable signal magnitudes even in deep networks. In Theorem 5.8, we combine this with SIC and normalization to obtain an explicit Lipschitz bound.

### 5.3. Gauge Equivariance and Phase Consistency

A central motivation for introducing complex-valued transport is to ensure the consistency under local rephasing of node embeddings. In standard message passing, scalar aggregation is sensitive to arbitrary sign or phase flips, which can destabilize attention and obscure directional relationships. By contrast, our formulation is gauge-equivariant: local phase transformations affect intermediate representations in a controlled, predictable way, and do not alter the final alignment scores or gating decisions.

**Theorem 5.5** (Gauge equivariance). *For any choice of node-wise phases $\{\psi_i\}_{i\in\mathcal{V}}$, consider the local $U(1)$ gauge transformation below:*

$$h_i^{(t)} \mapsto h_i^{\prime(t)} = e^{i\psi_i} h_i^{(t)}, \quad U_{ji} \mapsto U_{ji}^\prime = e^{i(\psi_i - \psi_j)} U_{ji}. \tag{37}$$

*For every head $m$ and edge $(j,i)$, the transported neighbor feature and the SIC residual transform covariantly as*

$$\tilde{h}_{j\to i}^{\prime(m)} = e^{i\psi_i} \tilde{h}_{j\to i}^{(m)}, \quad r_{j\to i}^{\prime(m)} = e^{i\psi_i} r_{j\to i}^{(m)}. \tag{38}$$

*Moreover, any scalar alignment score of the form*

$$s_{ji}^{(m)}(v) = \left(Q^{(m)} h_i^{(t)}\right)^{\mathrm{H}} v_{j\to i}^{(m)}, \tag{39}$$

*is gauge-invariant, i.e., $s_{ji}^{\prime(m)}(v^\prime) = s_{ji}^{(m)}(v)$ under transformation $v_{j\to i}^{(m)} \in \{\tilde{h}_{j\to i}^{(m)}, r_{j\to i}^{(m)}, \widehat{m}_{j\to i}^{(m)}\}$. Consequently, the GESC update is gauge-equivariant in the sense that*

$$\{h_i^{(t)}\}_{i\in\mathcal{V}} \mapsto \{e^{i\psi_i} h_i^{(t)}\}_{i\in\mathcal{V}} \tag{40}$$
$$\implies \{h_i^{(t+1)}\}_{i\in\mathcal{V}} \mapsto \{e^{i\psi_i} h_i^{(t+1)}\}_{i\in\mathcal{V}}.$$

*Proof is given in Appendix C.3.1.*

This gauge-equivariance property justifies the use of magnetic transport and complex embeddings: local phase choices at each node do not affect attention scores or gating decisions, ensuring robust and well-defined propagation across arbitrary local phase frames.

### 5.4. Stability and Non-amplification

We now show that the interference-aware design provides explicit stability guarantees, aligning with the gating and SIC mechanisms described in Section 4.

**Proposition 5.6** (Self-component non-amplification). *Consider the linear case (i.e., without nonlinearities or normal-*

*ization) with the per-layer update below:*

$$h_i^{(t+1)} = h_i^{(t)} + \sum_{m=1}^{M} \sum_{j\in\mathcal{N}(i)} \alpha_{ji}^{(m)} \widehat{m}_{j\to i}^{(m)}, \tag{41}$$

*where $\widehat{m}_{j\to i}^{(m)}$ is defined in Eq. 19. Then, for any node $i$,*

$$\left\|\Pi_\epsilon(h_i^{(t)}) h_i^{(t+1)}\right\|_2 \leq \left\|\Pi_\epsilon(h_i^{(t)}) h_i^{(t)}\right\|_2 \tag{42}$$
$$+ \sum_{m=1}^{M} \sum_{j\in\mathcal{N}(i)} \alpha_{ji}^{(m)} \left\|\Pi_\epsilon(h_i^{(t)}) \tilde{h}_{j\to i}^{(m)}\right\|_2.$$

*In particular, the SIC-processed residuals do not increase the self-parallel component. Their contribution in the direction of $h_i^{(t)}$ is always bounded above by that of the ungated transported messages, whose magnitude is controlled by the transported terms. Proof is in Appendix C.4.1.*

**Proposition 5.7** (Stability bound). *Let $\alpha_{\max} := \max_{m,i,j} \alpha_{ji}^{(m)}$, $\xi_{\max} := \max_{m,i,j} \xi_{ji}^{(m)}$, and assume the in-degree is bounded by $\Delta$ (i.e., $|\mathcal{N}(i)| \leq \Delta \ \forall\, i$). Then, for each head $m$ and node $i$, we can induce:*

$$\left\| \sum_{j\in\mathcal{N}(i)} \alpha_{ji}^{(m)} \xi_{ji}^{(m)} \left(g_{ji}^{(m)} r_{j\to i}^{(m)} + (1 - g_{ji}^{(m)}) \tilde{h}_{j\to i}^{(m)}\right) \right\|_2$$
$$\leq \alpha_{\max} \xi_{\max} \Delta \|W^{(m)}\|_2 \max_{j\in\mathcal{N}(i)} \|h_j^{(t)}\|_2. \tag{43}$$

*This shows that soft gating controls the per-layer amplification factor at the head level, ensuring bounded signal growth. Proof is given in Appendix C.4.2.*

**Theorem 5.8** (Lipschitz continuity and SIC-aware tightened bound). *Assume (i) NodeNorm and modReLU are non-expansive (each is 1-Lipschitz with respect to the $\ell_2$ norm), (ii) the magnetic transport $U_{ji}$ is unitary, and (iii) the SIC strength satisfies $0 \leq \eta_{\mathrm{sic}} \leq 1$ so that $\|(I - \eta_{\mathrm{sic}} \Pi_\epsilon(h_i^{(t)}))\|_2 \leq 1$ for all $i, t$. Let $\alpha_{ji}^{(m)} \leq \alpha_{\max}$ and suppose the in-degree is bounded by $\Delta$. The following bound applies to the coefficient-frozen propagation operator of one realized GESC layer, where attention weights, gates, and SIC operators are fixed after the forward pass.*

$$L_{\mathrm{GESC}} \leq 1 + \sum_{m=1}^{M} \alpha_{\max} \Delta \|W^{(m)}\|_2. \tag{44}$$

*Moreover, along the node-wise self-parallel direction $\mathrm{span}\{h_i^{(t)}\}$, there exist lower bounds $g_{\min}, \xi_{\min}, \eta_{\min} > 0$ on the applied gates and SIC strength such that*

$$L_{\mathrm{GESC}}^{\parallel} \leq 1 + \sum_{m=1}^{M} \alpha_{\max} \Delta \left(1 - g_{\min} \xi_{\min} \eta_{\min}\right) \|W^{(m)}\|_2, \tag{45}$$

*so that the directional Lipschitz factor along self-aligned directions is strictly reduced compared to the case without*

*Table 1.* (Q1) Node classification accuracy on nine benchmark datasets with **top-3 results**. The comparison includes heterophily-aware models, advanced spectral-filtering methods, and recent state-of-the-art architectures.

| Dataset | Cora | Citeseer | Pubmed | Actor | Chameleon | Squirrel | Cornell | Texas | Wisconsin |
|---|---|---|---|---|---|---|---|---|---|
| Homophily $\mathcal{G}_h$ (Eq. 1) | 0.81 | 0.74 | 0.80 | 0.22 | 0.23 | 0.22 | 0.11 | 0.06 | 0.16 |
| GCN (Kipf & Welling, 2017) | $81.4_{\pm0.71}$ | $67.5_{\pm0.70}$ | $79.5_{\pm0.47}$ | $20.3_{\pm0.46}$ | $54.9_{\pm0.59}$ | $31.1_{\pm0.71}$ | $39.9_{\pm0.79}$ | $57.0_{\pm0.90}$ | $49.0_{\pm0.78}$ |
| GAT (Veličković et al., 2018) | $82.6_{\pm0.55}$ | $68.4_{\pm0.83}$ | $79.9_{\pm0.45}$ | $22.8_{\pm0.41}$ | $54.4_{\pm0.84}$ | $31.0_{\pm0.93}$ | $42.6_{\pm0.80}$ | $58.8_{\pm1.01}$ | $50.2_{\pm0.97}$ |
| H₂GCN (Zhu et al., 2020) | $80.3_{\pm0.52}$ | $68.5_{\pm0.76}$ | $78.8_{\pm0.37}$ | $25.9_{\pm1.07}$ | $53.1_{\pm0.88}$ | $31.2_{\pm0.68}$ | $55.0_{\pm1.15}$ | $66.1_{\pm1.27}$ | $62.0_{\pm1.25}$ |
| SIGN (Frasca et al., 2020) | $82.4_{\pm0.57}$ | $68.6_{\pm0.70}$ | $79.7_{\pm0.43}$ | $22.6_{\pm0.40}$ | $54.2_{\pm0.80}$ | $31.2_{\pm0.04}$ | $42.4_{\pm0.85}$ | $58.8_{\pm0.95}$ | $50.4_{\pm0.88}$ |
| GCNII (Chen et al., 2020) | $82.2_{\pm0.64}$ | $67.8_{\pm1.21}$ | $79.4_{\pm0.52}$ | $26.2_{\pm1.22}$ | $54.0_{\pm0.77}$ | $30.8_{\pm0.91}$ | $56.0_{\pm1.27}$ | $69.1_{\pm1.34}$ | $63.9_{\pm1.29}$ |
| MagNet (Zhang et al., 2021) | $83.8_{\pm0.56}$ | $68.9_{\pm0.67}$ | $80.1_{\pm0.40}$ | $26.4_{\pm0.97}$ | $56.9_{\pm1.34}$ | $32.4_{\pm1.15}$ | $55.1_{\pm1.31}$ | $65.3_{\pm1.46}$ | $61.7_{\pm1.54}$ |
| GPRGNN (Chien et al., 2021) | $82.0_{\pm0.59}$ | $70.1_{\pm0.91}$ | $79.4_{\pm0.57}$ | $25.2_{\pm0.89}$ | $55.8_{\pm0.81}$ | $30.6_{\pm0.63}$ | $51.4_{\pm1.36}$ | $60.7_{\pm1.28}$ | $63.1_{\pm1.21}$ |
| FAGCN (Bo et al., 2021) | $82.8_{\pm0.66}$ | $70.4_{\pm1.20}$ | $79.8_{\pm0.55}$ | $26.8_{\pm1.24}$ | $54.8_{\pm0.81}$ | $31.2_{\pm0.87}$ | $56.8_{\pm1.22}$ | $69.7_{\pm1.41}$ | $64.3_{\pm1.25}$ |
| ACM-GCN (Luan et al., 2022) | $82.9_{\pm0.70}$ | $70.7_{\pm0.81}$ | $80.0_{\pm0.45}$ | $25.9_{\pm1.02}$ | $56.6_{\pm1.40}$ | $32.1_{\pm1.05}$ | $55.1_{\pm1.35}$ | $65.9_{\pm1.52}$ | $62.1_{\pm1.45}$ |
| GloGNN (Li et al., 2022a) | $82.9_{\pm0.45}$ | $70.9_{\pm0.48}$ | $\mathbf{80.3}_{\pm0.31}$ | $27.0_{\pm0.73}$ | $53.9_{\pm0.70}$ | $31.0_{\pm0.82}$ | $48.8_{\pm1.15}$ | $62.5_{\pm1.21}$ | $60.2_{\pm1.12}$ |
| Auto-HeG (Zheng et al., 2023) | $82.4_{\pm1.07}$ | $70.2_{\pm1.36}$ | $80.1_{\pm0.27}$ | $26.5_{\pm0.99}$ | $54.3_{\pm1.33}$ | $31.7_{\pm1.11}$ | $53.9_{\pm1.03}$ | $67.4_{\pm1.65}$ | $64.0_{\pm1.49}$ |
| DirGNN (Rossi et al., 2024) | $\mathbf{84.6}_{\pm0.61}$ | $\mathbf{71.5}_{\pm0.88}$ | $80.2_{\pm0.43}$ | $27.5_{\pm0.95}$ | $\mathbf{59.8}_{\pm1.45}$ | $35.2_{\pm1.13}$ | $\mathbf{57.9}_{\pm1.80}$ | $68.8_{\pm1.57}$ | $63.0_{\pm1.33}$ |
| PCNet (Li et al., 2024) | $83.4_{\pm0.77}$ | $70.8_{\pm1.15}$ | $80.0_{\pm0.29}$ | $26.6_{\pm0.90}$ | $57.6_{\pm1.65}$ | $31.8_{\pm0.58}$ | $54.1_{\pm1.02}$ | $62.5_{\pm1.16}$ | $60.5_{\pm1.13}$ |
| TFE-GNN (Duan et al., 2024) | $\mathbf{84.1}_{\pm0.72}$ | $\mathbf{71.7}_{\pm1.14}$ | $\mathbf{80.3}_{\pm0.30}$ | $\mathbf{28.1}_{\pm0.81}$ | $\mathbf{60.2}_{\pm1.61}$ | $\mathbf{36.0}_{\pm0.59}$ | $53.7_{\pm1.07}$ | $63.8_{\pm1.11}$ | $62.5_{\pm1.19}$ |
| CGNN (Zhuo et al., 2025) | $83.9_{\pm0.70}$ | $70.5_{\pm1.23}$ | $80.1_{\pm0.51}$ | $26.5_{\pm1.17}$ | $59.1_{\pm0.78}$ | $34.4_{\pm0.97}$ | $\mathbf{57.4}_{\pm1.25}$ | $\mathbf{70.3}_{\pm1.36}$ | $\mathbf{64.9}_{\pm1.22}$ |
| L2DGCN (Ding et al., 2025) | $82.4_{\pm0.99}$ | $71.5_{\pm0.47}$ | $79.8_{\pm0.20}$ | $\mathbf{31.3}_{\pm0.35}$ | $53.1_{\pm0.37}$ | $\mathbf{35.4}_{\pm0.52}$ | $51.5_{\pm3.28}$ | $\mathbf{76.7}_{\pm2.77}$ | $\mathbf{65.8}_{\pm3.01}$ |
| **GESC (ours)** | $\mathbf{84.9}_{\pm0.54}$ | $\mathbf{72.1}_{\pm0.51}$ | $\mathbf{80.4}_{\pm0.10}$ | $\mathbf{30.4}_{\pm0.66}$ | $\mathbf{65.0}_{\pm1.29}$ | $\mathbf{37.9}_{\pm0.36}$ | $\mathbf{59.4}_{\pm1.93}$ | $\mathbf{74.5}_{\pm1.52}$ | $\mathbf{66.6}_{\pm2.14}$ |

*SIC (i.e., $\eta_{\mathrm{sic}} = 0$). In particular, SIC does not worsen the worst-case Lipschitz constant and tightens it along self-aligned directions. Further discussion of over-smoothing mitigation and complete proofs is provided in Appendix C.5.*

# 6. Experiments

We conduct extensive experiments to evaluate our framework by addressing the following key questions:

- **Q1**: **Accuracy.** Does GESC improve node classification performance across both homophilous and heterophilous benchmarks?

- **Q2**: **Ablation Study.** How do SIC, sign-aware gating, and gauge-equivariant transport individually contribute to the overall performance?

- **Q3**: **Robustness and Sensitivity.** How sensitive is the model to variations in hyperparameters (interference cancellation rate $\eta_{\mathrm{sic}}$ and JS divergence $\lambda_{\mathrm{JS}}$) in Eq. 11 and 31, and how stable is it under edge perturbations?

- **Q4**: **Over-smoothing.** Do the proposed strategies alleviate over-smoothing and maintain stable signal propagation when stacking deeper layers?

The details of Datasets and Baselines are in Appendix D.

## 6.1. (Q1) Main Results

Table 1 reports the node classification accuracy on nine benchmarks. Classical propagation models such as GCN and GAT maintain competitive accuracy on strongly homophilic graphs (e.g., Cora and Citeseer), yet their effectiveness degrades sharply when edge-label correlation weakens

(e.g., Actor or Squirrel). Depth-regularized variants (GCNII and H₂GCN) alleviate over-smoothing but remain prone to overfitting on small-scale WebKB datasets. Methods that incorporate adaptive propagation or global re-weighting consistently outperform vanilla baselines on moderately heterophilic datasets (Cornell, Texas, Wisconsin), demonstrating the benefit of learned spectral filters and feedback aggregation. However, their advantage diminishes on WebKBs, where long-range spectral components dominate.

Spectral architectures (DirGNN, TFE-GNN, and CGNN) achieve substantial gains in these challenging settings by leveraging edge orientation and spectral calibration to preserve cross-community dependencies. Overall, the proposed GESC exhibits the most balanced and robust performance, ranking first on seven benchmarks and within the top three on the remaining two. Its consistent accuracy across both homophilic and heterophilic graphs demonstrates the effectiveness of the proposed self-interference cancellation and sign-aware gating mechanisms, which jointly harmonize local consistency and global structural information while maintaining low variance across random splits.

## 6.2. (Q2) Ablation Study

Table 2 presents the ablation results, where each major component of GESC is removed individually to assess its contribution. Removing the **Self-Interference Cancellation (w/o SIC)** significantly leads to lower accuracy across all benchmarks, verifying that eliminating self-aligned redundant signals before attention effectively mitigates over-smoothing and enhances feature diversity. The **Residual Gating (w/o RG)** mechanism also proves critical, as it adaptively balances constructive and destructive interference. Without RG, accuracy decreases by approximately 1.5-2.0% on average,

*Table 2.* (Q2) Ablation study on four benchmark datasets (Accuracy %). Each component is incrementally removed from the full GESC model to analyze its contribution.

| Variant | Cora | Citeseer | Chameleon | Texas |
|---|---|---|---|---|
| w/o SIC | 82.4 | 69.5 | 60.7 | 68.2 |
| w/o RG | 83.8 | 70.0 | 62.6 | 72.4 |
| w/o GET | 83.0 | 69.2 | 61.8 | 70.7 |
| w/o Complex | 82.6 | 68.6 | 60.9 | 70.1 |
| **Full GESC** | **84.9** | **72.1** | **65.0** | **74.5** |

indicating that the residual gate prevents the amplification of harmful correlations. The absence of **Gauge-Equivariant Transport (w/o GET)** causes one of the most pronounced performance drops (up to $-3.8\%$), demonstrating that phase-consistent message transport is vital for preserving robustness under sign and orientation perturbations. Finally, replacing complex-valued representations **with real-valued ones (w/o Complex)** consistently degrades performance, confirming that the coupled magnitude-phase representation provides richer and more expressive relational features than purely real-valued aggregation.

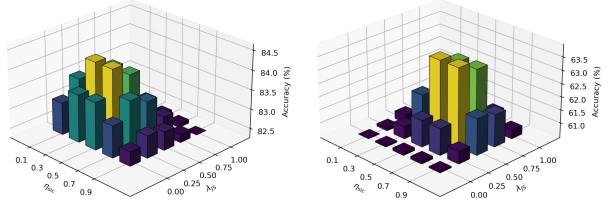

*(a)* Cora (homophilic)    *(b)* Chameleon (heterophilic)

*Figure 2.* (Q3) Node classification accuracy to $\eta_{\text{sic}}$ (Eq. 11) and $\lambda_{\text{JS}}$ (Eq. 31) on two datasets. The z-axis is normalized relative to the minimum accuracy for each graph.

### 6.3. (Q3) Robustness and Sensitivity

To assess the robustness of GESC, we analyze its sensitivity to two key hyperparameters: the information coupling coefficient $\eta_{\text{sic}}$ and the joint-smoothing regularizer $\lambda_{\text{JS}}$. Figure 2 illustrates the node classification accuracy across varying $(\eta_{\text{sic}}, \lambda_{\text{JS}})$ settings for the Cora and Chameleon datasets. The response surfaces vary smoothly over the tested grid and show a broad, high-performing region around the selected hyperparameters. Even when $\eta_{\text{sic}}$ or $\lambda_{\text{JS}}$ deviates from its optimal value, the performance degradation remains moderate (within 1–2%), demonstrating that the model maintains high accuracy without requiring precise hyperparameter tuning. This stability arises from the complementary effects of self-interference cancellation and residual gating, which jointly regulate the balance between local smoothing and global consistency. Overall, these results confirm that GESC

achieves robust generalization across diverse graph structures while exhibiting minimal sensitivity to hyperparameter selection.

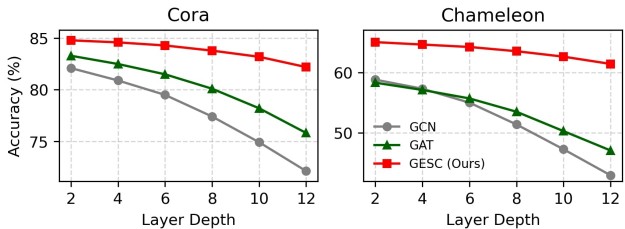

*Figure 3.* (Q4) Node classification accuracy versus network depth on the Cora (left) and Chameleon (right) datasets.

### 6.4. (Q4) Over-smoothing

To evaluate the effect of network depth, we analyze whether GESC mitigates the over-smoothing problem common in deep message-passing networks. As the number of layers increases, standard GCN and GAT models exhibit a sharp decline in accuracy due to feature homogenization across nodes, leading to indistinguishable embeddings. Figure 3 shows node classification accuracy on Cora and Chameleon when stacking up to 12 layers. In both datasets, GCN and GAT suffer progressive degradation with depth, confirming the accumulation of over-smoothed signals. In contrast, GESC maintains consistently higher accuracy, indicating that self-interference cancellation and residual gating effectively stabilize propagation and preserve discriminative information in deep architectures. This depth robustness arises from the interplay between local refinement and global consistency enforced by gauge-equivariant transport, preventing representation collapse as the network grows deeper.

## 7. Conclusion

We presented GESC, a gauge-equivariant graph neural network that replaces additive message passing with a principled wave-interference mechanism. The self-interference cancellation suppresses redundant self-reinforcement, while sign-aware gating balances constructive and destructive interference by aligning them. Theoretically, we show that the message-passing layers are gauge- and permutation-equivariant, and derive coefficient-frozen stability bounds indicating that SIC contracts self-parallel message components. Empirically, it achieves consistent improvements across both homophilous and heterophilous benchmarks, demonstrating robust and interpretable propagation dynamics. Beyond these results, GESC provides a geometric foundation for interference-aware graph learning, paving the way toward deeper, physically grounded models that unify topology, symmetry, and signal propagation.

## Acknowledgements

This research was supported by Sookmyung Women's University Research Grants (1-2503-2027), by the National Research Foundation of Korea (RS-2026-25468807), and by the Institute of Information & Communications Technology Planning & Evaluation (IITP)-Innovative Human Resource Development for Local Intellectualization program grant funded by the Korea government (MSIT) (RS-2026-25544527).

## Impact Statement

Potential benefits include more reliable node-level inference and better modeling of node interactions. GESC targets semi-supervised node classification under homophily/heterophily and is not a drop-in solution for safety-critical or privacy-sensitive tasks without additional safeguards. We do not foresee any specific negative societal consequences directly resulting from this work.

## Potential Limitations

First, GESC relies on complex-valued representations, which may introduce additional cost. Although the overall complexity remains linear in the number of edges, training can be sensitive to the nonlinearities. Second, the learned $U(1)$ transport captures phase relationships but does not directly extend to richer gauge groups. Finally, while GESC improves robustness under heterophily, its performance on extremely noisy graphs may still degrade.

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

| Model family | Method | Edge operators | Spectrum | Interference | Limitation |
|---|---|---|---|---|---|
| Homophily GNNs | GCN | Scalar adjacency | Implicit low-pass | None | Oversmoothing |
| | GAT | Learned attention | Implicit | None | No heterophily modeling |
| | GraphSAGE | Sampling-based | None | None | Limited spectral control |
| | JKNet | Skip connections | Implicit | None | Over-smoothing persists |
| | DropEdge | Edge dropout | Implicit | None | Stability issues |
| | GCNII | Residual | Flexible low-pass | None | Complexity increases |
| Heterophily GNNs | H2GCN | Decoupled features | High-frequency | Partial | Heuristic interference |
| | FAGCN | Signed filters | Adaptive spectral | Partial | No phase modeling |
| | ACM-GCN | Adaptive mixing | Multi-hop | Partial | Sensitive to noise |
| | MixHop | Hop mixing | Fixed spectrum | None | Limited adaptivity |
| | GBK-GNN | Gaussian kernels | Multi-band | None | Heavy tuning |
| | L2DGCN | Signed kernel | High-frequency | Partial | Instability on noise |
| Spectral GNNs | ChebNet | Polynomial filters | Spectral shaping | None | Fixed eigenbasis |
| | CayleyNet | Rational filters | Flexible spectrum | None | Requires eigendecomp |
| | BernNet | Bernstein basis | Smooth spectrum | None | High-order kernels |
| | GPR-GNN | Polynomial filters | Flexible response | None | No interference control |
| | JacobiConv | Jacobian filters | Spectral Jacobians | None | Linear approx only |
| Diffusion GNNs | SGC | Diffusion | Low-pass | None | Static filters |
| | SIGN | Multi-diffusion | Fixed filters | None | No adaptivity |
| | SignNet | Basis filters | Global structure | None | Rigid kernels |
| Geometric GNNs | NSD | Fixed sheaf maps | Sheaf Laplacian | Limited | No explicit cancellation |
| | SheafNN | Restrictions | Sheaf spectrum | Limited | Complex maps |
| | SheafAN | Sheaf attention | Sheaf filters | Limited | Superlinear cost |
| | NLSD | Nonlinear sheaf | Nonlinear spectrum | Limited | Hard to train |
| | Geom-GCN | Geometric operators | Geometric | Implicit | Weak interference control |
| Magnetic GNNs | MagNet | Magnetic Laplacian | Complex spectrum | Implicit only | Additive aggregation |
| | MSGNN | Magnetic Laplacian | Complex phases | Implicit | No cancellation |
| Gauge GNNs | GaugeCNN | Gauge-linear maps | Gauge-consistent | Implicit | No suppression of buildup |
| | **GESC (ours)** | **Learned U(1) + SIC** | **Hybrid phase** | **Explicit + sign** | **Controls interference** |

*Table 3.* Comparison across model families using grouping. Each method appears in its own row, with the corresponding family shown once on the left. GESC is the only model combining gauge-invariant transport with explicit interference cancellation. The reference of each model is introduced in the following section.

# A. Comparative Analysis

In this section, we provide a detailed comparison between the major GNN families summarized in Table 3. For each family, we discuss (i) the nature of its edge operators, (ii) the resulting spectral behavior, (iii) the presence or absence of interference modeling, and (iv) the structural limitations that arise from these choices. Then, we connect these observations to the mechanisms introduced in GESC, emphasizing how Self-Interference Cancellation (SIC), sign-aware gating, and gauge-consistent transport together address the core weaknesses of prior methods.

## A.1. Classical Homophily GNNs

Methods such as GCN (Kipf & Welling, 2017), GAT (Veličković et al., 2018), GraphSAGE (Hamilton et al., 2017), JKNet (Xu et al., 2018), DropEdge (Rong et al., 2019), and GCNII (Chen et al., 2020) rely on scalar adjacency information and adopt an additive neighborhood accumulation rule. Their propagation operators implicitly act as low-pass graph filters, repeatedly averaging node features with their neighbors. Because these models do not distinguish between self-parallel and orthogonal message components, the repeated reinforcement of the dominant low-frequency directions inevitably produces oversmoothing and loss of discriminative information. Even variants with skip connections or residual propagation preserve the same accumulation bias, as they lack any mechanism for attenuating redundant components or modeling phase-aligned interference. In contrast, GESC uses SIC to explicitly remove self-parallel redundancy before attention, thus mitigating the low-frequency collapse that characterizes this class.

## A.2. Heterophily-aware GNNs

Models targeting heterophily: H$_2$GCN (Zhu et al., 2020), FAGCN (Bo et al., 2021), ACM-GCN (Luan et al., 2022), MixHop (Abu-El-Haija et al., 2019), GBK-GNN (Du et al., 2022), and L2DGCN (Ding et al., 2025) introduce edge operators that amplify high-frequency signals or combine multi-hop neighborhoods. While these designs improve performance on

heterophilic graphs, they often rely on heuristics (e.g., signed filters or hop mixing) that provide only partial handling of interference. In particular, they lack an explicit mechanism for controlling when negatively aligned signals should be suppressed and when they carry useful information. This partial treatment causes instability in noisy settings and often requires extensive tuning. GESC directly addresses this gap: the sign-aware gate quantitatively modulates residual messages based on phase-consistent alignment, balancing suppression of harmful interference with preservation of informative high-frequency components.

## A.3. Spectral GNNs

Spectral methods such as ChebNet (Defferrard et al., 2016), CayleyNet (Levie et al., 2018), BernNet (He et al., 2021), GPR-GNN (Chien et al., 2021), and JacobiConv (Wang & Zhang, 2022) construct polynomial or rational filters tailored to specific spectral responses. While these techniques offer precise control over the eigenvalue domain, they operate in fixed or learned spectral bases without modeling local phase interactions between neighboring nodes. As a result, they lack a notion of interference during message passing, and the filtering operation remains purely magnitude-based. Moreover, many of these models require explicit eigendecomposition or produce high-order kernels that are expensive and difficult to tune. In contrast, GESC achieves spectral adaptivity implicitly through the SIC operator and complex-valued transport, generating phase-sensitive message adjustments without the need for eigenbasis computations.

## A.4. Global Diffusion GNNs

Global filtering approaches such as SGC (Wu et al., 2019a), SIGN (Frasca et al., 2020), and SignNet (Lim et al., 2022) remove learnable propagation entirely or replace it with a small number of global diffusion channels. These frameworks are computationally efficient but inherit a strong low-pass bias from their diffusion origins and lack any form of interference control. As a consequence, they are unable to separate harmful reinforcement (e.g., self-parallel buildup) from informative global structure. GESC, by contrast, preserves learnable and local edge-sensitive behavior while maintaining robustness through SIC and gauge-consistent modulation.

## A.5. Geometric and Sheaf-based GNNs

Geometric and sheaf-based methods: NSD (Bodnar et al., 2022), SheafNN (Hansen & Gebhart, 2020), SheafAN (Barbero et al., 2022), NLSD (Zaghen et al., 2024) operate on richer relational structures such as sheaf Laplacians or learned geometric operators. These approaches offer enhanced expressiveness but typically suffer from limited interference modeling and often incur substantial computational overhead. Their filters are defined in sheaf or geometric spectra, but do not distinguish constructive from destructive interference across neighbors. By contrast, GESC exploits a lightweight phase-based transport and sign-aware modulation that preserves geometric consistency while remaining scalable.

## A.6. Magnetic GNNs

MagNet (Zhang et al., 2021), MSGNN (He et al., 2022), and Geom-GCN (Pei et al., 2020) incorporate complex-valued transport, ensuring that messages transform consistently under local rephasing. While these models respect graph structure, they typically implement implicit-only interference handling: messages with incompatible phases are aggregated additively, with no explicit cancellation or sign-aware correction. As a result, interference may accumulate across layers and lead to unstable signal magnitudes. GESC extends this family by combining explicit SIC, sign-aware gating, and hybrid magnitude-phase attention, overcoming the buildup problem and improving robustness to noisy or misaligned neighbors.

**Why GESC is different?** Although GaugeCNN (Favoni et al., 2022) proposed gauge equivariant convolutional neural networks, it lacks interference cancellation and a hybrid view of messages. Comparatively, GESC is the first model to jointly incorporate: (i) Self-Interference Cancellation (SIC): a principled operator that removes redundant self-parallel components before attention, mitigating oversmoothing, (ii) sign-aware, phase-consistent gating: a gauge-invariant mechanism that suppresses harmful interference while preserving informative high-frequency contributions, and (iii) Gauge-consistent complex transport: ensuring that all alignment scores and gating decisions are invariant under local rephasing, stabilizing the propagation rule.

## B. Algorithmic and Implementation Details

### B.1. Overall Algorithm

---

**Algorithm 1** GESC: Gauge-Equivariant Self-Interference Cancellation (one training epoch)

---

**Require:** Graph $\mathcal{G} = (\mathcal{V}, \mathcal{E})$, real features $H_{\text{real}}^{(0)}$, labels $Y$; hyperparameters $\{L, M, \{\gamma_m\}_{m=1}^M, \lambda, \eta_{\text{sic}}, \epsilon, p_{\text{edge-drop}}, T, \lambda_{\text{JS}}\}$
**Ensure:** Updated parameters $\Theta$

1: **Complex lift:** $H^{(0)} \leftarrow H_{\text{real}}^{(0)} W_{\text{re}}^{(0)} + i\, H_{\text{real}}^{(0)} W_{\text{im}}^{(0)}$
2: **Magnetic transports:** For $(j \to i) \in \mathcal{E}$, set $U_{ji} = e^{i\theta_{ji}}$ (unit-modulus)
3: **for** $\ell = 1$ to $L$ **do**
4:    $H^{(\ell)} \leftarrow 0$ {accumulator (same shape as $H^{(\ell-1)}$)}
5:    **for** $m = 1$ to $M$ **do**
6:       **for all** $(j \to i) \in \mathcal{E}$ **do**
7:          **Source transport and SIC projector:** $\tilde{h}_{j \to i}^{(m)} \leftarrow U_{ji} W^{(m)} h_j^{(\ell-1)}$,    $\Pi_\epsilon(h_i^{(\ell-1)}) \leftarrow \dfrac{h_i^{(\ell-1)}(h_i^{(\ell-1)})^H}{\|h_i^{(\ell-1)}\|_2^2 + \epsilon}$
8:          **Self-interference cancellation:** $r_{j \to i}^{(m)} \leftarrow \tilde{h}_{j \to i}^{(m)} - \eta_{\text{sic}} \cdot \Pi_\epsilon(h_i^{(\ell-1)}) \tilde{h}_{j \to i}^{(m)}$
9:          **Pre-gate complex score:** $s_{ji}^{(m)} \leftarrow (Q^{(m)} h_i^{(\ell-1)})^H r_{j \to i}^{(m)}$
10:         **Gate normalizer:** $\nu_{ji,\text{gate}}^{(m)} \leftarrow \|Q^{(m)} h_i^{(\ell-1)}\|_2 \cdot \|r_{j \to i}^{(m)}\|_2 + \epsilon$
11:         **Phase-consistent alignment:** $\rho_{ji}^{(m)} \leftarrow \text{Re}\left( \dfrac{s_{ji}^{(m)}}{\nu_{ji,\text{gate}}^{(m)}} \right)$
12:         **Sign-aware gate and residual:** $\xi_{ji}^{(m)} \leftarrow \sigma\left(c_m \rho_{ji}^{(m)} + d_m\right)$,    $\bar{r}_{j \to i}^{(m)} \leftarrow \xi_{ji}^{(m)} r_{j \to i}^{(m)}$
13:         **Residual gate:** $x_{ji}^{(m)} \leftarrow \left[\log(1 + \|\bar{r}_{j \to i}^{(m)}\|_2),\ \log(1 + \|\tilde{h}_{j \to i}^{(m)}\|_2),\ \log(1 + |s_{ji}^{(m)}|)\right]$,    $g_{ji}^{(m)} \leftarrow \sigma(a_m^\top x_{ji}^{(m)})$
14:         **Post-gate message and score:** $\widehat{m}_{j \to i}^{(m)} \leftarrow g_{ji}^{(m)} \bar{r}_{j \to i}^{(m)} + \left(1 - g_{ji}^{(m)}\right) \tilde{h}_{j \to i}^{(m)}$,    $\tilde{s}_{ji}^{(m)} \leftarrow (Q^{(m)} h_i^{(\ell-1)})^H \widehat{m}_{j \to i}^{(m)}$
15:         **Attention normalizer:** $\nu_{ji,\text{attn}}^{(m)} \leftarrow \|Q^{(m)} h_i^{(\ell-1)}\|_2 \cdot \|\widehat{m}_{j \to i}^{(m)}\|_2 + \epsilon$
16:         **Hybrid attention logit:** $\ell_{ji}^{(m)} \leftarrow \gamma_m \left[ \lambda \dfrac{|\tilde{s}_{ji}^{(m)}|}{\sqrt{d}} + (1 - \lambda) \text{Re}\left( \dfrac{\tilde{s}_{ji}^{(m)}}{\nu_{ji,\text{attn}}^{(m)}} \right) \right]$
17:       **end for**
18:       **for all** $i \in \mathcal{V}$ **do**
19:          **Attention:** $\alpha_{ji}^{(m)} \leftarrow \text{softmax}_{j \in \mathcal{N}(i)}\left(\ell_{ji}^{(m)}\right)$
20:       **end for**
21:       **for all** $(j \to i) \in \mathcal{E}$ **do**
22:          **Aggregate (head-$m$):** $H_i^{(\ell)} \mathrel{+}= \alpha_{ji}^{(m)} \widehat{m}_{j \to i}^{(m)}$
23:       **end for**
24:    **end for**
25:    **Residual update:** $\widetilde{h}_i^{(\ell)} \leftarrow h_i^{(\ell-1)} + H_i^{(\ell)}$    $\forall i$
26:    **Node-wise stats (on $\widetilde{h}_i^{(\ell)}$):** $\mu_i \leftarrow \frac{1}{d} \sum_k \widetilde{h}_{ik}^{(\ell)}$,   $\sigma_i \leftarrow \sqrt{\frac{1}{d} \sum_k |\widetilde{h}_{ik}^{(\ell)} - \mu_i|^2}$
27:    **NodeNorm and Complex activation:** $\text{NN}(i) \leftarrow (\widetilde{h}_i^{(\ell)} - \mu_i)/(\sigma_i + \epsilon)$,    $h_i^{(\ell)} \leftarrow \text{modReLU}\big(\text{NN}(i)\big)$    $\forall i$
28: **end for**
29: **Readout (real classifier):**
30:    $z_i \leftarrow \text{vec}\big(\text{Re}(h_i^{(L)}), \text{Im}(h_i^{(L)})\big)$;    $\text{logits}_i \leftarrow W_{\text{cls}} \text{LayerNorm}\big(\text{Dropout}(z_i)\big)$
31: **Cross-entropy:** $\mathcal{L}_{\text{CE}} \leftarrow -\frac{1}{|\mathcal{V}_L|} \sum_{i \in \mathcal{V}_L} \log p_\Theta(y_i \mid \text{logits}_i)$
32: **JS consistency with edge dropout:**
33:    Sample two independent edge-drop masks with prob $p_{\text{edge-drop}}$ and re-run forward to get $\text{logits}^{(1)}, \text{logits}^{(2)}$
34:    $\mathcal{L}_{\text{JS}} \leftarrow \text{JS}\left(\text{softmax}\big(\text{logits}^{(1)}/T\big), \text{softmax}\big(\text{logits}^{(2)}/T\big)\right)$
35: **Total loss:** $\mathcal{L} \leftarrow \mathcal{L}_{\text{CE}} + \lambda_{\text{JS}} \mathcal{L}_{\text{JS}}$
36: **Update:** $\Theta \leftarrow \text{OptimizerStep}\big(\Theta, \nabla_\Theta \mathcal{L}\big)$

---

## B.2. Implementation Details

All experiments are implemented in PyTorch Geometric with complex-valued extensions for message passing. We train all models using the Adam optimizer with a learning rate of $1 \times 10^{-3}$ and weight decay $5 \times 10^{-4}$. A single Titan XP GPU is used for all experiments. Following (Kipf & Welling, 2017), we use 20 labeled nodes per class for training and randomly split the remaining nodes into validation and test sets (50%/50%). Early stopping is applied with patience 100 based on validation accuracy. All experiments are repeated over 10 random seeds to report the mean and standard deviation.

**Model structure and propagation.** Each GESC layer consists of (i) complex linear projection, (ii) gauge-equivariant transport, (iii) self-interference cancellation (SIC), and (iv) sign-aware gating with magnitude-phase attention. The transport $U_{ji}$ is represented as a phase vector and applied through elementwise complex multiplication during edge aggregation. The SIC projection is implemented as a Tikhonov-regularized rank-1 projection

$$P_i^{\perp} = I - \eta_{\text{sic}} \Pi_{\epsilon}(h_i), \tag{46}$$

computed via a single complex inner product and outer product per edge. This operation is fused with message construction to avoid redundant memory access. Gating and attention weights are computed in parallel from real-valued scalar magnitudes and normalized with degree-wise softmax.

**Sparse and parallel computation.** All message passing operations are performed with sparse gather/scatter primitives. Phase transport and SIC are vectorized over edges to minimize per-edge kernel launches, and multi-head attention is parallelized across heads using fused CUDA kernels. The resulting computational complexity scales linearly with the number of edges $E$ up to the $d^2$ term from complex projections, as analyzed in Section 4.3.

# C. Theoretical Proof

## C.1. Why SIC Works: a Linear-Algebraic View

### C.1.1. PROOF OF PROPOSITION 5.1

Let $h = h_i^{(t)}$. If $h = 0$, then $\Pi_{\epsilon}(h) = 0$ and the SIC operator is the identity. The claim is then trivial under the zero self-parallel convention. We consider $h \neq 0$. Define

$$P_h := \frac{h h^{\text{H}}}{\|h\|_2^2}, \quad \lambda_h := \frac{\|h\|_2^2}{\|h\|_2^2 + \epsilon} \in (0, 1). \tag{47}$$

Then $P_h$ is the orthogonal rank-1 projector onto $\text{span}\{h\}$ and

$$\Pi_{\epsilon}(h) = \lambda_h P_h. \tag{48}$$

The decomposition $\tilde{h} = \tilde{h}_{\|} + \tilde{h}_{\perp}$ satisfies $P_h \tilde{h}_{\|} = \tilde{h}_{\|}$ and $P_h \tilde{h}_{\perp} = 0$. Therefore,

$$(I - \eta_{\text{sic}} \Pi_{\epsilon}(h))\tilde{h} = (I - \eta_{\text{sic}} \lambda_h P_h)(\tilde{h}_{\|} + \tilde{h}_{\perp}) \tag{49}$$

$$= (1 - \eta_{\text{sic}} \lambda_h)\tilde{h}_{\|} + \tilde{h}_{\perp} \tag{50}$$

$$= \left(1 - \eta_{\text{sic}} \frac{\|h_i^{(t)}\|_2^2}{\|h_i^{(t)}\|_2^2 + \epsilon}\right)\tilde{h}_{\|} + \tilde{h}_{\perp}. \tag{51}$$

Thus, SIC attenuates the self-parallel component and preserves the orthogonal component. $\square$

### C.1.2. PROOF OF LEMMA 5.2

Let $h = h_i^{(t)}$. If $h = 0$, then $\Pi_{\epsilon}(h) = 0$ and both sides of the desired inequality are zero. We consider $h \neq 0$ and set

$$\lambda_h := \frac{\|h\|_2^2}{\|h\|_2^2 + \epsilon} \in (0, 1). \tag{52}$$

The Tikhonov-regularized operator satisfies

$$\Pi_{\epsilon}(h) = \lambda_h P_h, \quad \Pi_{\epsilon}(h)^2 = \lambda_h \Pi_{\epsilon}(h), \tag{53}$$

where $P_h$ is the orthogonal rank-1 projector onto $\mathrm{span}\{h\}$. For any $\tilde{h} \in \mathbb{C}^d$,

$$\Pi_\epsilon(h)(I - \eta_{\mathrm{sic}}\Pi_\epsilon(h))\tilde{h} = \Pi_\epsilon(h)\tilde{h} - \eta_{\mathrm{sic}}\Pi_\epsilon(h)^2\tilde{h} \tag{54}$$

$$= (1 - \eta_{\mathrm{sic}}\lambda_h)\Pi_\epsilon(h)\tilde{h}. \tag{55}$$

Consequently,

$$\mathcal{E}_\|\big((I - \eta_{\mathrm{sic}}\Pi_\epsilon(h))\tilde{h}\big) = (1 - \eta_{\mathrm{sic}}\lambda_h)^2\mathcal{E}_\|(\tilde{h}). \tag{56}$$

Since $0 \le \eta_{\mathrm{sic}} \le 1$ and $0 < \lambda_h < 1$, we have $0 \le 1 - \eta_{\mathrm{sic}}\lambda_h \le 1$. Thus,

$$\mathcal{E}_\|\big((I - \eta_{\mathrm{sic}}\Pi_\epsilon(h))\tilde{h}\big) \le \mathcal{E}_\|(\tilde{h}), \tag{57}$$

which proves that SIC does not increase the squared self-parallel energy measured by $\Pi_\epsilon(h)$. $\square$

### C.1.3. DISCUSSION FOR REMARK 5.3

Let $h^{(t)} = \sum_k c_k^{(t)} u_k$ be the expansion of node features in the Laplacian eigenbasis $\{u_k\}$. For normalized adjacency $\tilde{A}$, suppose

$$\tilde{A}u_k = \mu_k u_k, \tag{58}$$

with eigenvalues $\mu_k \in [-1, 1]$, where low-frequency modes correspond to larger $\mu_k$ under diffusion-like propagation. Then,

$$h^{(t+1)} \approx \alpha h^{(t)} + \tilde{A}h^{(t)} \tag{59}$$

yields

$$c_k^{(t+1)} \approx (\alpha + \mu_k)c_k^{(t)}, \tag{60}$$

so the low-frequency coefficients are preferentially amplified when $\alpha + \mu_k$ is larger. Applying SIC introduces the local contraction

$$I - \eta_{\mathrm{sic}}\Pi_\epsilon(h_i^{(t)}). \tag{61}$$

By Proposition 5.1, this contraction suppresses the component parallel to the current local state $h_i^{(t)}$. When repeated diffusion has aligned local states with dominant low-frequency components, SIC attenuates those locally dominant directions before attention. This gives a node-wise notch-filter interpretation and explains why SIC can delay spectral collapse. The argument is a local spectral interpretation, not a global eigenvalue theorem for the nonlinear layer. $\square$

## C.2. Stability from Sign-aware Gating

### C.2.1. PROOF OF PROPOSITION 5.4

Fix a head $m$ and a target node $i$. If $\mathcal{N}(i) = \varnothing$, the neighbor aggregation is the zero vector under the zero aggregation convention. We therefore consider $\mathcal{N}(i) \neq \varnothing$. Recall that

$$\tilde{h}_{j\to i}^{(m)} = U_{ji}W^{(m)}h_j^{(t)}, \tag{62}$$

where $U_{ji}$ is unitary. Since $U_{ji}$ has unit modulus,

$$\|\tilde{h}_{j\to i}^{(m)}\|_2 = \|W^{(m)}h_j^{(t)}\|_2 \le \|W^{(m)}\|_2\|h_j^{(t)}\|_2. \tag{63}$$

The operator

$$\Pi_\epsilon(h_i^{(t)}) = \frac{h_i^{(t)}(h_i^{(t)})^{\mathrm{H}}}{\|h_i^{(t)}\|_2^2 + \epsilon} \tag{64}$$

is Hermitian positive semidefinite and has spectrum

$$\left\{\frac{\|h_i^{(t)}\|_2^2}{\|h_i^{(t)}\|_2^2 + \epsilon}, 0, \ldots, 0\right\}. \tag{65}$$

Thus, $\|\Pi_\epsilon(h_i^{(t)})\|_2 \leq 1$. For $0 \leq \eta_{\text{sic}} \leq 1$, the eigenvalues of $I - \eta_{\text{sic}}\Pi_\epsilon(h_i^{(t)})$ lie in $[0, 1]$, and therefore

$$\|I - \eta_{\text{sic}}\Pi_\epsilon(h_i^{(t)})\|_2 \leq 1. \tag{66}$$

It follows that

$$\|r_{j\to i}^{(m)}\|_2 = \|(I - \eta_{\text{sic}}\Pi_\epsilon(h_i^{(t)}))\tilde{h}_{j\to i}^{(m)}\|_2 \leq \|\tilde{h}_{j\to i}^{(m)}\|_2. \tag{67}$$

Since $0 \leq \xi_{ji}^{(m)} \leq 1$,

$$\|\bar{r}_{j\to i}^{(m)}\|_2 = \|\xi_{ji}^{(m)} r_{j\to i}^{(m)}\|_2 \leq \|\tilde{h}_{j\to i}^{(m)}\|_2. \tag{68}$$

The post-gate message is a convex combination of $\bar{r}_{j\to i}^{(m)}$ and $\tilde{h}_{j\to i}^{(m)}$:

$$\widehat{m}_{j\to i}^{(m)} = g_{ji}^{(m)} \bar{r}_{j\to i}^{(m)} + (1 - g_{ji}^{(m)})\tilde{h}_{j\to i}^{(m)}. \tag{69}$$

Consequently, by the triangle inequality and $0 \leq g_{ji}^{(m)} \leq 1$,

$$\|\widehat{m}_{j\to i}^{(m)}\|_2 \leq g_{ji}^{(m)}\|\bar{r}_{j\to i}^{(m)}\|_2 + (1 - g_{ji}^{(m)})\|\tilde{h}_{j\to i}^{(m)}\|_2 \leq \|\tilde{h}_{j\to i}^{(m)}\|_2. \tag{70}$$

Using $\alpha_{ji}^{(m)} \geq 0$ and $\sum_{j\in\mathcal{N}(i)} \alpha_{ji}^{(m)} = 1$,

$$\left\| \sum_{j\in\mathcal{N}(i)} \alpha_{ji}^{(m)} \widehat{m}_{j\to i}^{(m)} \right\|_2 \leq \sum_{j\in\mathcal{N}(i)} \alpha_{ji}^{(m)} \|\widehat{m}_{j\to i}^{(m)}\|_2 \tag{71}$$

$$\leq \max_{j\in\mathcal{N}(i)} \|\tilde{h}_{j\to i}^{(m)}\|_2 \tag{72}$$

$$\leq \|W^{(m)}\|_2 \max_{j\in\mathcal{N}(i)} \|h_j^{(t)}\|_2. \tag{73}$$

This proves the claim. $\square$

## C.3. Gauge Equivariance and Phase Consistency

### C.3.1. PROOF OF THEOREM 5.5

Let local $U(1)$ gauge transformations act as

$$h_i^{(t)} \mapsto h_i'^{(t)} = e^{i\psi_i}h_i^{(t)}, \quad U_{ji} \mapsto U_{ji}' = e^{i(\psi_i - \psi_j)}U_{ji}. \tag{74}$$

Assume $W^{(m)}$ and $Q^{(m)}$ act on the channel dimension and are fixed under the local phase action. Since $e^{i\psi_i}$ is a scalar, it commutes with these complex-linear maps. The transported neighbor feature transforms as

$$\tilde{h}_{j\to i}'^{(m)} = U_{ji}' W^{(m)} h_j'^{(t)} \tag{75}$$

$$= e^{i(\psi_i - \psi_j)} U_{ji} W^{(m)} e^{i\psi_j} h_j^{(t)} \tag{76}$$

$$= e^{i\psi_i} \tilde{h}_{j\to i}^{(m)}. \tag{77}$$

The regularized rank-1 operator is invariant under the target phase:

$$\Pi_\epsilon(h_i'^{(t)}) = \frac{(e^{i\psi_i}h_i^{(t)})(e^{i\psi_i}h_i^{(t)})^{\text{H}}}{\|e^{i\psi_i}h_i^{(t)}\|_2^2 + \epsilon} \tag{78}$$

$$= \frac{h_i^{(t)}(h_i^{(t)})^{\text{H}}}{\|h_i^{(t)}\|_2^2 + \epsilon} = \Pi_\epsilon(h_i^{(t)}). \tag{79}$$

Consequently,

$$r_{j\to i}'^{(m)} = \tilde{h}_{j\to i}'^{(m)} - \eta_{\text{sic}}\Pi_\epsilon(h_i'^{(t)})\tilde{h}_{j\to i}'^{(m)} \tag{80}$$

$$= e^{i\psi_i}\left(\tilde{h}_{j\to i}^{(m)} - \eta_{\text{sic}}\Pi_\epsilon(h_i^{(t)})\tilde{h}_{j\to i}^{(m)}\right) \tag{81}$$

$$= e^{i\psi_i} r_{j\to i}^{(m)}. \tag{82}$$

The target-filtered feature satisfies

$$Q^{(m)}h_i'^{(t)} = e^{i\psi_i}Q^{(m)}h_i^{(t)}. \tag{83}$$

Thus, if $v_{j\to i}'^{(m)} = e^{i\psi_i}v_{j\to i}^{(m)}$, then

$$s_{ji}'^{(m)}(v') = \left(Q^{(m)}h_i'^{(t)}\right)^{\mathrm{H}}v_{j\to i}'^{(m)} \tag{84}$$

$$= \left(e^{i\psi_i}Q^{(m)}h_i^{(t)}\right)^{\mathrm{H}}e^{i\psi_i}v_{j\to i}^{(m)} \tag{85}$$

$$= s_{ji}^{(m)}(v). \tag{86}$$

This proves gauge invariance of all scalar scores computed from transported or SIC-residual messages. Norms, magnitudes, normalized real alignments, and logarithmic magnitude features are also invariant under multiplication by $e^{i\psi_i}$. Therefore, the scalar gates $\xi_{ji}^{(m)}$ and $g_{ji}^{(m)}$ are gauge-invariant.

Since $\xi_{ji}^{(m)}$ is invariant,

$$\bar{r}_{j\to i}'^{(m)} = \xi_{ji}^{(m)}r_{j\to i}'^{(m)} = e^{i\psi_i}\bar{r}_{j\to i}^{(m)}. \tag{87}$$

Since $g_{ji}^{(m)}$ is invariant, the post-gate message satisfies

$$\widehat{m}_{j\to i}'^{(m)} = g_{ji}^{(m)}\bar{r}_{j\to i}'^{(m)} + (1 - g_{ji}^{(m)})\tilde{h}_{j\to i}'^{(m)} \tag{88}$$

$$= e^{i\psi_i}\widehat{m}_{j\to i}^{(m)}. \tag{89}$$

The post-gate score is therefore invariant:

$$\left(Q^{(m)}h_i'^{(t)}\right)^{\mathrm{H}}\widehat{m}_{j\to i}'^{(m)} = \left(Q^{(m)}h_i^{(t)}\right)^{\mathrm{H}}\widehat{m}_{j\to i}^{(m)}. \tag{90}$$

The attention logits use only this invariant score, its magnitude, its normalized real part, and invariant norms. Thus, $\ell_{ji}^{(m)}$ and $\alpha_{ji}^{(m)}$ are gauge-invariant. The pre-activation update transforms as

$$\widetilde{h}_i'^{(t+1)} = h_i'^{(t)} + \sum_{m=1}^{M}\sum_{j\in\mathcal{N}(i)}\alpha_{ji}^{(m)}\widehat{m}_{j\to i}'^{(m)} \tag{91}$$

$$= e^{i\psi_i}\left(h_i^{(t)} + \sum_{m=1}^{M}\sum_{j\in\mathcal{N}(i)}\alpha_{ji}^{(m)}\widehat{m}_{j\to i}^{(m)}\right) \tag{92}$$

$$= e^{i\psi_i}\widetilde{h}_i^{(t+1)}. \tag{93}$$

Finally, NodeNorm and modReLU are phase-equivariant. For any $z \in \mathbb{C}^d$ and any scalar phase $e^{i\psi}$,

$$\mathrm{NodeNorm}(e^{i\psi}z) = e^{i\psi}\mathrm{NodeNorm}(z), \quad \mathrm{modReLU}(e^{i\psi}z) = e^{i\psi}\mathrm{modReLU}(z), \tag{94}$$

because the node-wise mean transforms by the same phase, while centered magnitudes and variances remain unchanged. Thus, the hidden GESC layer update is gauge-equivariant. $\square$

### C.3.2. PERMUTATION EQUIVARIANCE OF THE UPDATE

Let $\pi$ be any permutation of nodes and let $P_\pi$ be the corresponding permutation matrix. Relabel node features, edges, and transports by

$$h_i^{(t)} \mapsto h_{\pi(i)}'^{(t)} = h_i^{(t)}, \quad U_{ji} \mapsto U_{\pi(j)\pi(i)}' = U_{ji}. \tag{95}$$

For every edge $(j, i)$ and head $m$, the transported message after relabeling satisfies

$$\tilde{h}_{\pi(j)\to\pi(i)}'^{(m)} = U_{\pi(j)\pi(i)}'W^{(m)}h_{\pi(j)}'^{(t)} = \tilde{h}_{j\to i}^{(m)}. \tag{96}$$

The same identity holds for $\Pi_\epsilon(h_i^{(t)})$, $r_{j\to i}^{(m)}$, $s_{ji}^{(m)}$, $\xi_{ji}^{(m)}$, $g_{ji}^{(m)}$, $\widehat{m}_{j\to i}^{(m)}$, and $\ell_{ji}^{(m)}$ after replacing $(j,i)$ with $(\pi(j),\pi(i))$. The neighbor set of $\pi(i)$ in the relabeled graph is exactly $\{\pi(j) : j \in \mathcal{N}(i)\}$. Since the softmax is applied over the same multiset of logits,

$$\alpha'^{(m)}_{\pi(j)\pi(i)} = \alpha_{ji}^{(m)}. \tag{97}$$

Therefore,

$$\widetilde{h}'^{(t+1)}_{\pi(i)} = h'^{(t)}_{\pi(i)} + \sum_{m=1}^{M} \sum_{\pi(j)\in\mathcal{N}(\pi(i))} \alpha'^{(m)}_{\pi(j)\pi(i)} \widehat{m}'^{(m)}_{\pi(j)\to\pi(i)} \tag{98}$$

$$= h_i^{(t)} + \sum_{m=1}^{M} \sum_{j\in\mathcal{N}(i)} \alpha_{ji}^{(m)} \widehat{m}_{j\to i}^{(m)} \tag{99}$$

$$= \widetilde{h}_i^{(t+1)}. \tag{100}$$

NodeNorm and modReLU are applied independently at each node, so they preserve this relabeling relation. Thus, the layer is permutation-equivariant. $\square$

### C.4. Stability and Non-amplification

#### C.4.1. PROOF OF PROPOSITION 5.6

We consider the linearized layer update

$$h_i^{(t+1)} = h_i^{(t)} + \sum_{m=1}^{M} \sum_{j\in\mathcal{N}(i)} \alpha_{ji}^{(m)} \widehat{m}_{j\to i}^{(m)}, \tag{101}$$

where

$$\widehat{m}_{j\to i}^{(m)} = g_{ji}^{(m)} \xi_{ji}^{(m)} r_{j\to i}^{(m)} + (1 - g_{ji}^{(m)}) \tilde{h}_{j\to i}^{(m)}, \quad r_{j\to i}^{(m)} = (I - \eta_{\text{sic}} \Pi_\epsilon(h_i^{(t)})) \tilde{h}_{j\to i}^{(m)}. \tag{102}$$

Applying $\Pi_\epsilon(h_i^{(t)})$ and using the triangle inequality gives

$$\left\|\Pi_\epsilon(h_i^{(t)}) h_i^{(t+1)}\right\|_2 \leq \left\|\Pi_\epsilon(h_i^{(t)}) h_i^{(t)}\right\|_2 \tag{103}$$

$$+ \sum_{m=1}^{M} \sum_{j\in\mathcal{N}(i)} \alpha_{ji}^{(m)} \left\|\Pi_\epsilon(h_i^{(t)}) \widehat{m}_{j\to i}^{(m)}\right\|_2. \tag{104}$$

For each edge and head,

$$\left\|\Pi_\epsilon(h_i^{(t)}) \widehat{m}_{j\to i}^{(m)}\right\|_2 \leq g_{ji}^{(m)} \xi_{ji}^{(m)} \left\|\Pi_\epsilon(h_i^{(t)}) r_{j\to i}^{(m)}\right\|_2 \tag{105}$$

$$+ (1 - g_{ji}^{(m)}) \left\|\Pi_\epsilon(h_i^{(t)}) \tilde{h}_{j\to i}^{(m)}\right\|_2. \tag{106}$$

By Lemma 5.2,

$$\left\|\Pi_\epsilon(h_i^{(t)}) r_{j\to i}^{(m)}\right\|_2 \leq \left\|\Pi_\epsilon(h_i^{(t)}) \tilde{h}_{j\to i}^{(m)}\right\|_2. \tag{107}$$

Since $0 \leq g_{ji}^{(m)}, \xi_{ji}^{(m)} \leq 1$,

$$g_{ji}^{(m)} \xi_{ji}^{(m)} + 1 - g_{ji}^{(m)} = 1 - g_{ji}^{(m)}(1 - \xi_{ji}^{(m)}) \leq 1. \tag{108}$$

Combining these bounds yields

$$\left\|\Pi_\epsilon(h_i^{(t)}) h_i^{(t+1)}\right\|_2 \leq \left\|\Pi_\epsilon(h_i^{(t)}) h_i^{(t)}\right\|_2 \tag{109}$$

$$+ \sum_{m=1}^{M} \sum_{j\in\mathcal{N}(i)} \alpha_{ji}^{(m)} \left\|\Pi_\epsilon(h_i^{(t)}) \tilde{h}_{j\to i}^{(m)}\right\|_2. \tag{110}$$

Thus, SIC-processed residuals do not contribute more self-parallel magnitude than their uncancelled transported messages. $\square$

### C.4.2. PROOF OF PROPOSITION 5.7

Fix a head $m$ and node $i$. If $\mathcal{N}(i) = \varnothing$, the neighbor aggregation is the zero vector under the zero aggregation convention. We therefore consider $\mathcal{N}(i) \neq \varnothing$. For the exact post-gate message in Eq. 19,

$$\widehat{m}_{j \to i}^{(m)} = g_{ji}^{(m)} \xi_{ji}^{(m)} r_{j \to i}^{(m)} + (1 - g_{ji}^{(m)}) \tilde{h}_{j \to i}^{(m)}. \tag{111}$$

By unitary transport and the contraction property of SIC,

$$\|r_{j \to i}^{(m)}\|_2 \leq \|\tilde{h}_{j \to i}^{(m)}\|_2 \leq \|W^{(m)}\|_2 \|h_j^{(t)}\|_2. \tag{112}$$

Therefore,

$$\|\widehat{m}_{j \to i}^{(m)}\|_2 \leq g_{ji}^{(m)} \xi_{ji}^{(m)} \|r_{j \to i}^{(m)}\|_2 + (1 - g_{ji}^{(m)}) \|\tilde{h}_{j \to i}^{(m)}\|_2 \tag{113}$$

$$\leq \|W^{(m)}\|_2 \|h_j^{(t)}\|_2. \tag{114}$$

Thus, the implemented per-head aggregation obeys

$$\Big\| \sum_{j \in \mathcal{N}(i)} \alpha_{ji}^{(m)} \widehat{m}_{j \to i}^{(m)} \Big\|_2 \leq \alpha_{\max} \Delta \|W^{(m)}\|_2 \max_{j \in \mathcal{N}(i)} \|h_j^{(t)}\|_2. \tag{115}$$

The displayed quantity in Proposition 5.7 is the gate-scaled comparison aggregation

$$S_i^{(m)} := \sum_{j \in \mathcal{N}(i)} \alpha_{ji}^{(m)} \xi_{ji}^{(m)} \big( g_{ji}^{(m)} r_{j \to i}^{(m)} + (1 - g_{ji}^{(m)}) \tilde{h}_{j \to i}^{(m)} \big). \tag{116}$$

Let

$$u_{j \to i}^{(m)} := g_{ji}^{(m)} r_{j \to i}^{(m)} + (1 - g_{ji}^{(m)}) \tilde{h}_{j \to i}^{(m)}. \tag{117}$$

Since $0 \leq g_{ji}^{(m)} \leq 1$,

$$\|u_{j \to i}^{(m)}\|_2 \leq g_{ji}^{(m)} \|r_{j \to i}^{(m)}\|_2 + (1 - g_{ji}^{(m)}) \|\tilde{h}_{j \to i}^{(m)}\|_2 \tag{118}$$

$$\leq \|W^{(m)}\|_2 \|h_j^{(t)}\|_2. \tag{119}$$

Using $\alpha_{ji}^{(m)} \leq \alpha_{\max}$, $\xi_{ji}^{(m)} \leq \xi_{\max}$, and $|\mathcal{N}(i)| \leq \Delta$, we obtain

$$\|S_i^{(m)}\|_2 \leq \sum_{j \in \mathcal{N}(i)} \alpha_{ji}^{(m)} \xi_{ji}^{(m)} \|u_{j \to i}^{(m)}\|_2 \tag{120}$$

$$\leq \alpha_{\max} \xi_{\max} \Delta \|W^{(m)}\|_2 \max_{j \in \mathcal{N}(i)} \|h_j^{(t)}\|_2. \tag{121}$$

This proves the displayed stability bound and records the corresponding bound for the exact implemented aggregation. $\square$

### C.4.3. PROOF OF THEOREM 5.8

We prove the coefficient-frozen Lipschitz bound associated with one realized forward pass of a GESC layer. In this bound, the attention weights, scalar gates, and SIC operators are fixed after being computed at the current layer state. This gives the propagation stability constant displayed in Theorem 5.8. If the dependence of these coefficients on the input features is also differentiated, then additional terms from the Tikhonov projector, sigmoid gates, normalized scores, and softmax attention appear. On bounded feature sets these coefficient terms are finite because $\epsilon > 0$, the graph degree is finite, and all learned parameters are finite. Thus, the displayed constant is the frozen-coefficient propagation part of the layer Lipschitz bound.

Let $\|H\|_{2,\infty} := \max_i \|h_i\|_2$ denote the node-wise block norm. Let $\delta h_j$ be a perturbation of $h_j^{(t)}$. For any edge $(j, i)$ and head $m$,

$$\delta \tilde{h}_{j \to i}^{(m)} = U_{ji} W^{(m)} \delta h_j. \tag{122}$$

Since $U_{ji}$ is unitary,

$$\|\delta \tilde{h}_{j\to i}^{(m)}\|_2 \leq \|W^{(m)}\|_2 \|\delta h_j\|_2. \tag{123}$$

For the frozen SIC operator,

$$\delta r_{j\to i}^{(m)} = (I - \eta_{\text{sic}} \Pi_\epsilon(h_i^{(t)})) \delta \tilde{h}_{j\to i}^{(m)}. \tag{124}$$

Assumption (iii) gives

$$\|\delta r_{j\to i}^{(m)}\|_2 \leq \|\delta \tilde{h}_{j\to i}^{(m)}\|_2 \leq \|W^{(m)}\|_2 \|\delta h_j\|_2. \tag{125}$$

With fixed gates $\xi_{ji}^{(m)}, g_{ji}^{(m)} \in [0,1]$, the perturbation of the post-gate message is

$$\delta \widehat{m}_{j\to i}^{(m)} = g_{ji}^{(m)} \xi_{ji}^{(m)} \delta r_{j\to i}^{(m)} + (1 - g_{ji}^{(m)}) \delta \tilde{h}_{j\to i}^{(m)}. \tag{126}$$

Using Eq. 125,

$$\|\delta \widehat{m}_{j\to i}^{(m)}\|_2 \leq \|W^{(m)}\|_2 \|\delta h_j\|_2. \tag{127}$$

The frozen-coefficient pre-activation perturbation is

$$\delta \widetilde{h}_i^{(t+1)} = \delta h_i^{(t)} + \sum_{m=1}^M \sum_{j \in \mathcal{N}(i)} \alpha_{ji}^{(m)} \delta \widehat{m}_{j\to i}^{(m)}. \tag{128}$$

Taking norms and using $\alpha_{ji}^{(m)} \leq \alpha_{\max}$ and $|\mathcal{N}(i)| \leq \Delta$ gives

$$\|\delta \widetilde{h}_i^{(t+1)}\|_2 \leq \|\delta h_i^{(t)}\|_2 + \sum_{m=1}^M \sum_{j \in \mathcal{N}(i)} \alpha_{ji}^{(m)} \|\delta \widehat{m}_{j\to i}^{(m)}\|_2 \tag{129}$$

$$\leq \left(1 + \sum_{m=1}^M \alpha_{\max} \Delta \|W^{(m)}\|_2\right) \|\delta H^{(t)}\|_{2,\infty}. \tag{130}$$

Taking the maximum over $i$ yields

$$\|\delta \widetilde{H}^{(t+1)}\|_{2,\infty} \leq \left(1 + \sum_{m=1}^M \alpha_{\max} \Delta \|W^{(m)}\|_2\right) \|\delta H^{(t)}\|_{2,\infty}. \tag{131}$$

By Assumption (i), NodeNorm and modReLU are non-expansive, so applying them after the pre-activation update does not increase the bound. Therefore, for the frozen-coefficient propagation operator,

$$L_{\text{GESC}} \leq 1 + \sum_{m=1}^M \alpha_{\max} \Delta \|W^{(m)}\|_2. \tag{132}$$

When the per-head softmax normalization is used directly, $\sum_j \alpha_{ji}^{(m)} = 1$ gives the sharper per-head convex aggregation bound with $\alpha_{\max} \Delta$ replaced by 1.

We next prove the SIC-aware self-parallel bound under the same frozen-coefficient convention. Decompose

$$\delta \tilde{h}_{j\to i}^{(m)} = \delta \tilde{h}_{\|}^{(m)} + \delta \tilde{h}_\perp^{(m)} \tag{133}$$

along $\text{span}\{h_i^{(t)}\}$ and its orthogonal complement. By Proposition 5.1,

$$(I - \eta_{\text{sic}} \Pi_\epsilon(h_i^{(t)})) \delta \tilde{h}_{j\to i}^{(m)} = (1 - \eta_{\text{sic}} \lambda_i^{(t)}) \delta \tilde{h}_{\|}^{(m)} + \delta \tilde{h}_\perp^{(m)}, \tag{134}$$

where

$$\lambda_i^{(t)} = \frac{\|h_i^{(t)}\|_2^2}{\|h_i^{(t)}\|_2^2 + \epsilon}. \tag{135}$$

The self-parallel part of $\delta\widehat{m}_{j\to i}^{(m)}$ is scaled by

$$g_{ji}^{(m)}\xi_{ji}^{(m)}(1-\eta_{\mathrm{sic}}\lambda_i^{(t)})+(1-g_{ji}^{(m)}) = 1 - g_{ji}^{(m)}(1-\xi_{ji}^{(m)}) - g_{ji}^{(m)}\xi_{ji}^{(m)}\eta_{\mathrm{sic}}\lambda_i^{(t)} \tag{136}$$

$$\leq 1 - g_{ji}^{(m)}\xi_{ji}^{(m)}\eta_{\mathrm{sic}}\lambda_i^{(t)}. \tag{137}$$

Let $g_{\min}$ and $\xi_{\min}$ be positive lower bounds on the gates over the considered realized layer, and let $\eta_{\min}$ be a positive lower bound on the effective SIC factor $\eta_{\mathrm{sic}}\lambda_i^{(t)}$. If this effective lower bound is zero, the directional result reduces to the global frozen-coefficient bound. Otherwise,

$$\left\|\Pi_\epsilon(h_i^{(t)})\delta\widehat{m}_{j\to i}^{(m)}\right\|_2 \leq (1-g_{\min}\xi_{\min}\eta_{\min})\left\|\Pi_\epsilon(h_i^{(t)})\delta\tilde{h}_{j\to i}^{(m)}\right\|_2. \tag{138}$$

Aggregating over neighbors and heads gives

$$L_{\mathrm{GESC}}^{\|} \leq 1 + \sum_{m=1}^{M}\alpha_{\max}\Delta(1-g_{\min}\xi_{\min}\eta_{\min})\|W^{(m)}\|_2. \tag{139}$$

Thus, SIC does not increase the worst-case frozen-coefficient propagation bound and strictly tightens the self-parallel bound whenever $g_{\min}\xi_{\min}\eta_{\min} > 0$. $\square$

## C.5. Over-smoothing mitigation

Consider the coefficient-frozen propagation operator of a GESC layer composed of Self-Interference Cancellation (SIC), sign-aware gating, and gauge-equivariant transport. Under the conditions of Propositions 5.6 and 5.7, and Theorem 5.8, the following consequences hold.

By Proposition 5.6, the component of each node state aligned with its previous representation obeys the bound

$$\left\|\Pi_\epsilon(h_i^{(t)})h_i^{(t+1)}\right\|_2 \leq \left\|\Pi_\epsilon(h_i^{(t)})h_i^{(t)}\right\|_2 + \sum_{m=1}^{M}\sum_{j\in\mathcal{N}(i)}\alpha_{ji}^{(m)}\left\|\Pi_\epsilon(h_i^{(t)})\tilde{h}_{j\to i}^{(m)}\right\|_2. \tag{140}$$

In particular, since SIC contracts the self-parallel part of each transported message, the residual branch does not contribute more self-parallel magnitude than the corresponding uncancelled transported message. This controls reinforcement of self-aligned directions that can lead to feature homogenization.

From Proposition 5.7, the magnitude of the gate-scaled comparison aggregation for each head $m$ and node $i$ is bounded by

$$\left\|\sum_{j\in\mathcal{N}(i)}\alpha_{ji}^{(m)}\xi_{ji}^{(m)}\big(g_{ji}^{(m)}r_{j\to i}^{(m)}+(1-g_{ji}^{(m)})\tilde{h}_{j\to i}^{(m)}\big)\right\|_2 \leq \alpha_{\max}\xi_{\max}\Delta\|W^{(m)}\|_2\max_{j\in\mathcal{N}(i)}\|h_j^{(t)}\|_2. \tag{141}$$

For the exact post-gate message in Eq. 19, the implemented aggregation satisfies the companion bound

$$\left\|\sum_{j\in\mathcal{N}(i)}\alpha_{ji}^{(m)}\widehat{m}_{j\to i}^{(m)}\right\|_2 \leq \alpha_{\max}\Delta\|W^{(m)}\|_2\max_{j\in\mathcal{N}(i)}\|h_j^{(t)}\|_2. \tag{142}$$

Thus, each head contributes a bounded update whose growth is controlled by the operator norm of $W^{(m)}$ and the maximal in-degree.

By Theorem 5.8, the coefficient-frozen propagation operator of a GESC layer satisfies

$$L_{\mathrm{GESC}} \leq 1 + \sum_{m=1}^{M}\alpha_{\max}\Delta\|W^{(m)}\|_2, \tag{143}$$

and along the self-parallel subspace,

$$L_{\mathrm{GESC}}^{\|} \leq 1 + \sum_{m=1}^{M}\alpha_{\max}\Delta(1-g_{\min}\xi_{\min}\eta_{\min})\|W^{(m)}\|_2, \tag{144}$$

*Table 4.* Statistics of the nine graph datasets.

| Datasets | Cora | Citeseer | Pubmed | Actor | Chameleon | Squirrel | Cornell | Texas | Wisconsin |
|---|---|---|---|---|---|---|---|---|---|
| Nodes | 2,708 | 3,327 | 19,717 | 7,600 | 2,277 | 5,201 | 183 | 183 | 251 |
| Edges | 10,558 | 9,104 | 88,648 | 25,944 | 33,824 | 211,872 | 295 | 309 | 499 |
| Features | 1,433 | 3,703 | 500 | 931 | 2,325 | 2,089 | 1,703 | 1,703 | 1,703 |
| Classes | 7 | 6 | 3 | 5 | 5 | 5 | 5 | 5 | 5 |

for positive lower bounds $g_{\min}, \xi_{\min}, \eta_{\min}$ on the gates and the effective SIC factor over the considered realized layer. The factor $(1 - g_{\min}\xi_{\min}\eta_{\min}) < 1$ shows that the effective amplification along self-aligned directions is reduced compared with propagation without SIC. In this sense, SIC and sign-aware gating tighten the stability bound precisely where over-smoothing is most likely to occur.

**Spectral implication (heuristic).** Let $H^{(t)}$ denote the node embedding matrix and $\mathcal{L}$ the graph Laplacian with eigen-decomposition $\mathcal{L} = U\Lambda U^\top$. Under diffusion-like propagation, low-frequency modes dominate since $\tilde{A}u_k \approx (1 - \lambda_k)u_k$ with small $\lambda_k$. The SIC and sign-aware gating mechanisms attenuate components aligned with locally dominant directions $h_i^{(t)}$, which, in diffusion regimes, tend to correlate with low-frequency eigenmodes. Heuristically, one can interpret a GESC layer as inducing a frequency-wise damping of the form

$$\|U^\top H^{(t+1)}\|_F^2 \approx \sum_k (1 - c_{\text{sic}}(k))^2 \|U^\top H^{(t)}\|_{F,k}^2, \tag{145}$$

where $c_{\text{sic}}(k)$ increases with the effective SIC strength $g_{\min}\xi_{\min}\eta_{\min}$ and the local alignment of mode $u_k$ with the node-wise directions. This suggests that locally dominant low-frequency energy is damped while higher-frequency structurally discriminative modes can be preserved. This spectral statement is heuristic and follows from the local contraction bounds, not from a global diagonalization of the nonlinear GESC operator.

**Summary.** Together, the above bounds indicate that GESC mitigates spectral collapse and over-smoothing in the coefficient-frozen propagation sense: the self-parallel component is controlled relative to uncancelled transported messages, the per-layer update norm remains bounded, and locally dominant low-frequency modes are selectively suppressed by SIC and sign-aware gating, leading to stable and diversity-preserving propagation even in deep stacks.

## D. Details of Datasets and Baselines

**Datasets.** To comprehensively evaluate model performance under varying structural regimes, we conduct experiments on nine widely adopted benchmark datasets, covering both homophilic and heterophilic graphs that are described in Table 4. The datasets include three classical citation networks: Cora, Citeseer, and Pubmed (Kipf & Welling, 2017), characterized by high homophily ($\mathcal{G}_h \in [0.74, 0.81]$), and six additional benchmarks that exhibit strong heterophily: Actor, Chameleon, Squirrel, Cornell, Texas, and Wisconsin (Rozemberczki et al., 2019). Actor is a co-occurrence network derived from film actor collaborations, while Chameleon and Squirrel are Wikipedia page networks with low homophily and rich structural diversity. Cornell, Texas, and Wisconsin are WebKB networks with extremely low $\mathcal{G}_h$ (0.06-0.16), often used as canonical heterophily benchmarks. We follow the same data splits and preprocessing protocols as prior heterophily literature (Zhu et al., 2020; Bo et al., 2021; Li et al., 2022a), including fixed 20 labeled nodes per class for training and the remaining nodes split evenly into validation and test sets.

**Baselines.** To ensure fair and comprehensive comparison, we benchmark our method against 15 representative GNN baselines that cover classical, heterophily-aware, spectral-filtering, and recent state-of-the-art methods.

- **Classical GNNs:** GCN (Kipf & Welling, 2017) and GAT (Veličković et al., 2018) are the standard message-passing architectures, serving as strong baselines in homophilic settings.

- **Heterophily-aware propagation:** H$_2$GCN (Zhu et al., 2020) separates ego and neighbor aggregation to alleviate bias, GPRGNN (Chien et al., 2021) learns personalized propagation weights, and FAGCN (Bo et al., 2021) adaptively mixes low- and high-frequency signals.

- **Spectral and signed filtering:** SIGN (Frasca et al., 2020) precomputes propagation features, MagNet (Zhang et al., 2021) introduces magnetic Laplacians for directional structure, GCNII (Chen et al., 2020) incorporates identity mapping to counter over-smoothing, and L2DGCN (Ding et al., 2025) mitigates bias-degree via learnable graph augmentation.

- **Adaptive and structure-enhanced models:** ACM-GCN (Luan et al., 2022) uses channel mixing, GloGNN (Li et al., 2022a) adds global nodes to enhance long-range propagation, Auto-HeG (Zheng et al., 2023) automates heterophily architecture search, DirGNN (Rossi et al., 2024) explicitly models directed edges, PCNet (Li et al., 2024) filters homophilic signals in heterophilic graphs, TFE-GNN (Duan et al., 2024) integrates topology and feature filtering, and CGNN (Zhuo et al., 2025) uses contrastive objectives to improve generalization.

This diverse set of baselines allows us to systematically compare GESC with (i) early message-passing GNNs, (ii) heterophily-specialized models, and (iii) recent spectral and structural advances, evaluating its effectiveness across a wide spectrum of graph conditions.

