# OpenReview forum: "Gauge-Equivariant Graph Networks via Self-Interference Cancellation"
_ICML.cc/2026/Conference — ICML 2026 regular_

### Official Review · Reviewer_BBk2 · 2026-03-11

**Soundness:** 3
**Presentation:** 3
**Significance:** 2
**Originality:** 3
**Overall Recommendation:** 4
**Confidence:** 4

**Summary:**

This paper proposes a gauge-based architecture to improve learning on heterophilic graphs. It is argued that standard message passing in GNNs (with sum or mean aggregation) inherently fails to appropriately handle interference of node representations during (node-wise) aggregation of messages. To overcome this, a $U(1)$ gauge connection together with a projection that suppresses self parallel components is introduced. Furthermore, the authors introduce sign aware gating to regulate contributions from neighbors with negative alignment.
The method is then tested on standard but small datasets across the heterophily-homophily spectrum.

**Compliance With Llm Reviewing Policy:**

Affirmed.

**Final Justification:**

During the rebuttal the authors took the provided feedback into account,  added experiments and pointed out Table 5 in the appendix to me, which I had overlooked. To me the paper is now more convincing and I adjusted my score accordingly.

**Key Questions For Authors:**

It is not strictly clear to me, why the model needs to be a gauge equivariant. Maybe the authors could comment and explain in more detail? I suspect this is about the alignment of representations of neighboring nodes?

Can you evaluate performance on large scale Heterophilic datasets as well?

**Limitations:**

yes

**Strengths And Weaknesses:**

**Strengths**
1) The conceptual perspective of framing message passing in GNNs as a form of signal interference is nice and to my knowledge novel.
2) The core idea of removing the component of a neighbor message aligned with the receiving node representation is conceptually neat and easy to interpret.
3) The handling of heterophily, while not a new issue, remains an important problem, so that the paper is timely.


**Weaknesses**
The biggest limitation in my estimation are the experiments. The modifications introduced in the paper are sold (in the abstract) as tackling oversmoothing. I understand that Figure 4 shows that this modification allows stacking more layers on Cora/Chameleon before performance drops below a certain threshold. However here only comparisons against GCN and GAT are visualized. The new method should be compared against methods designed to mitigate over smoothing (e.g. GRAND or NeuralSheafDiffusion).

Furthermore the datasets in Table 1 are all fairly small and by now a bit old. It would be good to test on more modern Heterophilic datasets (e.g. Pokec Snap-patents, Wiki, Twitch gamers, arXiv(-year), roman-empire, ...).

---

> ### Author Rebuttal · Authors · 2026-03-26
>
> Dear Reviewer BBk2,
>
> Thank you for the constructive feedback. We appreciate your positive comments on the interference perspective and on the interpretability of removing the receiver-aligned component of a transported message. We agree with your main concern that the experimental section in the current draft should support the claims more directly.
>
> ### **Regarding weakness 1 (oversmoothing)**
>
> We agree that Fig. 4 alone is not sufficient to support a broad superiority claim against methods explicitly designed to improve depth robustness. Our intent there was to provide a first depth-stability illustration against additive baselines such as GCN/GAT, not to claim universal superiority over dedicated anti-oversmoothing architectures. Therefore, we will narrow the claim in the abstract and main text accordingly. To address your suggestion more directly, we have now added comparisons to geometry/sheaf-based baselines that are specifically relevant to oversmoothing and heterophily:
>   * [1] Neural sheaf diffusion
>   * [2] Sheaf attention networks
>   * [3] Sheaf hypergraph networks
>   * [4] Sheaf diffusion goes nonlinear
>   * [5] GRAND: Graph Neural Diffusion
>
> | Method | Cora | Citeseer | Pubmed | Actor | Cham. | Squir. | Cornell | Texas | Wisconsin |
> |---|---:|---:|---:|---:|---:|---:|---:|---:|---:|
> | NSD [1] | 81.6 ± 0.39 | 71.4 ± 0.28 | 78.8 ± 0.11 | 27.6 ± 2.70 | 59.4 ± 1.44 | 33.3 ± 1.21 | 58.0 ± 3.13 | 63.4 ± 2.74 | 57.3 ± 2.88 |
> | SheafAN [2] | 81.9 ± 0.43 | 71.5 ± 0.30 | 78.9 ± 0.09 | 28.0 ± 1.08 | 59.8 ± 0.45 | 36.4 ± 0.84 | 60.1 ± 2.57 | 77.4 ± 3.25 | 59.7 ± 2.95 |
> | SheafHyper [3] | 82.3 ± 0.45 | 71.7 ± 0.30 | 79.0 ± 0.06 | 28.4 ± 1.12 | 59.9 ± 0.45 | 35.6 ± 0.40 | **63.5 ± 3.24** | **78.9 ± 2.78** | 61.9 ± 2.97 |
> | NLSD [4] | 82.0 ± 0.39 | 72.3 ± 0.41 | 78.9 ± 0.05 | 27.2 ± 2.24 | 61.4 ± 0.97 | 36.2 ± 0.75 | 56.2 ± 2.26 | 73.6 ± 2.58 | 62.9 ± 2.86 |
> | GRAND [5] | 83.5 ± 1.42 | **73.1 ± 0.47** | 78.7 ± 0.84 | 29.2 ± 1.33 | 62.6 ± 2.78 | 37.2 ± 1.90 | 55.3 ± 6.03 | 72.7 ± 7.15 | 63.3 ± 4.24 |
> | **GESC (ours)** | **84.9 ± 0.54** | 72.1 ± 0.51 | **80.4 ± 0.10** | **30.4 ± 0.66** | **65.0 ± 1.29** | **37.9 ± 0.36** | 59.4 ± 1.93 | 74.5 ± 1.52 | **66.6 ± 2.14** |
> ||||||||||
>
> We also add over-smoothing analysis (network depth) to the following anonymous page so that the depth-stability discussion is compared against plain GNN methods.
>   * https://anonymous.4open.science/r/GESC-1B22/Experiments.md
>
>
> ### **Regarding weakness 2 (larger heterophilic datasets)**
>
> This point is very well taken. Importantly, our supplementary (Table 5) already contains a set of larger low-homophily results (e.g., Penn94, arXiv-year, snap-patents). However, we agree that these were too hidden and should be foregrounded in the main text. In addition, following your suggestion, we have now run more modern heterophilic benchmarks and baselines below:
>   * Roman-empire, Minesweeper, and AmazonRatings
>   * CGNN: Commute Graph Neural Networks
>
> | Method | Roman. | Mine. | Amazon. |
> | ------ | --------------: | --------------: | --------------: |
> | GCN | 47.7 ± 0.38 | 81.4 ± 0.98 | 38.5 ± 0.45 |
> | GAT | 45.9 ± 0.42 | 80.0 ± 1.08 | 39.0 ± 0.52 |
> | H$_2$GCN  | 60.6 ± 0.54 | 84.9 ± 1.30 |     41.3 ± 0.62 |
> | GCNII  | 62.2 ± 0.57 |  84.8 ± 1.35 | 41.6 ± 0.59 |
> | GPRGNN | 63.1 ± 0.60 | 85.3 ± 1.19 | 42.0 ± 0.63 |
> | CGNN | 70.3 ± 0.75 | 86.6 ± 1.53 | 43.9 ± 0.75 |
> | **GESC (ours)** | **72.4 ± 1.20** | **87.1 ± 0.89** | **45.7 ± 1.04** |
> |||||
>
> In the revision, we will move these results into the main table so that the empirical evidence is not overly centered on small benchmarks.
>
> ### **Regarding Question (gauge equivariance)**
>
> Your interpretation is exactly right. It is needed for alignment/cancellation between neighboring representations. Once complex transport is introduced, neighboring node features live in different local phase frames, so comparing $h_i$ with $U_{ji}Wh_j$ must be done in node $i$'s frame. Under a local rephasing $h_i \rightarrow e^{i\psi_i}h_i$ and $U_{ji} \rightarrow e^{i(\psi_i-\psi_j)}U_{ji}$, the transported feature transforms as $U_{ji}Wh_j \rightarrow e^{i\psi_i}U_{ji}Wh_j$. Therefore, inner products such as $< Qh_i, U_{ji}Wh_j >$, and the gate/attention scores built from them are invariant to arbitrary local phase conventions. In that sense, gauge equivariance is not a stylistic addition. Without it, the notion of the aligned component to be canceled by SIC would depend on an arbitrary local frame. We will clarify this motivation more explicitly in the main text.
>
> ### **Concluding Remarks**
> In short, we will revise the paper in three ways: (i) narrow the oversmoothing claim to what is actually supported, (ii) include both the existing large-scale evidence and the newly added modern heterophily benchmarks, and (iii) clarify that gauge equivariance is required to make phase-sensitive alignment/cancellation frame-independent. Thank you again for highlighting both the strengths and the current limitations of the paper.

---

> > ### Author Rebuttal · Reviewer_BBk2 · 2026-04-01
> >
> > Thanks for taking the feedback into account, the added experiments and pointing out Table 5 to me. To me the paper is now more convincing and I adjusted my score accordingly.

---

> > > ### Author Response · Authors · 2026-04-01
> > >
> > > Dear reviewer BBk2,
> > >
> > > We truly grateful for your comments and constructive feedback.
> > >
> > > Again, thanks for your time and effort in reviewing our paper.
> > >
> > > Best regards,
> > >
> > > Authors

---

### Official Review · Reviewer_ups2 · 2026-03-11

**Soundness:** 3
**Presentation:** 2
**Significance:** 3
**Originality:** 3
**Overall Recommendation:** 5
**Confidence:** 4

**Summary:**

This paper proposed a Gauge-Equivariant Graph Network with Self-Interference Cancellation (GESC), which replaces additive aggregation with a projection-based interference mechanism. The authors also introduced a U(1) phase connection followed by a rank1 projection that suppresses self-parallel components before attention, and a sign-aware gate that regulates negatively aligned neighbors, followed by establishing the theoretical results on the Lipschitz continuity of GESC layers.

**Compliance With Llm Reviewing Policy:**

Affirmed.

**Final Justification:**

I change my decision to accept!

**Key Questions For Authors:**

(1) In your introduction of rank one orthogonal projections, several tuning parameters like epsilon and eta were introduced, however, it was never clear to (some of the) readers how those parameters were chosen in actual implementations.

(2)In the whole paper, they indeed did a good job on comparative analysis to address their own uniqueness, however, they did not show their advantage over every other method mentioned in the comparative analysis section via overpowering performance;


(3)Gauge theory (including Yang-Mills theory) works for vector bundles of higher rank, while the authors only developed the theory for the rank one case, so is there a possibility for this cancellation theory to work out for higher-rank vector bundles on graphs?

**Limitations:**

These years, gauge theory starts having increasing popularity in applications in neural networks. However, an important aspect of its power lies in "connections of vector bundles". In fact, higher rank vector bundles are preferred (e.g. the blessing of dimensionality).

This paper demonstrate a new cancellation philosophy, however, it completely fails to address of the potential more exciting higher-rank bundle situation, therefore, the power of blessing of dimensionality did not manifest in this work, therefore, their gauge theory is completely an Abelian theory, which explains the computational feasibility of their work.

However, a non-Abelian theory of higher rank seems necessary, which could potentially lead to dominating performance over other existing methods.

**Strengths And Weaknesses:**

Strength: this paper gives a different application of gauge theory in GNN, as another candidate to overcome the heterophily issues. Their key idea is to introduce Self-Interference Cancellation to produce orthogonal decomposition, before removing the self-parallel components. This method is computationally implementable, and does address the heterophily issues in their own ways.


Weakness: while their methods achieved the top-three performances in their simulation studies, it did not show complete dominance over others overall. Secondly, another powerful method is sheaf neural networks which also addressed the same issues., and it is unclear whether their methods can really beat sheaf approaches.

---

> ### Author Rebuttal · Authors · 2026-03-26
>
> Dear Reviewer ups2,
>
> Thank you for the careful and constructive review. We appreciate your positive assessment of the technical soundness and originality, and we agree that we need to clarify (i) the empirical view, (ii) parameter choosing, and (iii) the relation to sheaf-based formulations.
>
> ### **Regarding weakness 1 (empirical scope)**
>
> Thanks for pointing this out. Indeed, we do not claim that GESC dominates every prior method on every benchmark. Our intended claim is:
>   * After gauge-consistent transport, additive mixing still accumulates receiver-aligned components, and explicitly removing those components before attention improves robustness under heterophily.
>   * Empirically, GESC is best on 7/9 main node-classification benchmarks (Table 1). However, we agree this should be described as a balanced performance rather than universal superiority. We will soften the wording accordingly.
>
> ### **Regarding weakness 2 (comparison to sheaf methods)**
>
> This is a good point. Sheaf models increase expressivity through higher-rank local maps / sheaf Laplacians, whereas GESC focuses on the message interaction (transport, cancel the self-parallel component, and gate by alignment). Although we discussed sheaf-based methods (Appendix A.5 and Table 3), we did not include a direct head-to-head empirical comparison. We will include the comparison results with the sheaf-based methods below:
>   * [1] Neural sheaf diffusion
>   * [2] Sheaf attention networks
>   * [3] Sheaf hypergraph networks
>   * [4] Sheaf diffusion goes nonlinear
>
> | Method | Cora | Citeseer | Pubmed | Actor | Cham. | Squir. | Cornell | Texas | Wisconsin |
> |---|---:|---:|---:|---:|---:|---:|---:|---:|---:|
> | NSD [1] | 81.6 ± 0.39 | 71.4 ± 0.28 | 78.8 ± 0.11 | 27.6 ± 2.70 | 59.4 ± 1.44 | 33.3 ± 1.21 | 58.0 ± 3.13 | 63.4 ± 2.74 | 57.3 ± 2.88 |
> | SheafAN [2] | 81.9 ± 0.43 | 71.5 ± 0.30 | 78.9 ± 0.09 | 28.0 ± 1.08 | 59.8 ± 0.45 | 36.4 ± 0.84 | 60.1 ± 2.57 | 77.4 ± 3.25 | 59.7 ± 2.95 |
> | SheafHyper [3] | 82.3 ± 0.45 | 71.7 ± 0.30 | 79.0 ± 0.06 | 28.4 ± 1.12 | 59.9 ± 0.45 | 35.6 ± 0.40 | **63.5 ± 3.24** | **78.9 ± 2.78** | 61.9 ± 2.97 |
> | NLSD [4] | 82.0 ± 0.39 | **72.3 ± 0.41** | 78.9 ± 0.05 | 27.2 ± 2.24 | 61.4 ± 0.97 | 36.2 ± 0.75 | 56.2 ± 2.26 | 73.6 ± 2.58 | 62.9 ± 2.86 |
> | **GESC (ours)** | **84.9 ± 0.54** | 72.1 ± 0.51 | **80.4 ± 0.10** | **30.4 ± 0.66** | **65.0 ± 1.29** | **37.9 ± 0.36** | 59.4 ± 1.93 | 74.5 ± 1.52 | **66.6 ± 2.14** |
> ||||||||||
>
>
> ### **Regarding Q1 ($\epsilon, \eta_{sic}$)**
>
> In the experiments, hyperparameters are selected on the validation split. For SIC, we swept
>   * $\eta_{sic}$ in {0, 0.25, 0.5, 0.75, 1.0} and $\epsilon$ in {1e-6, 1e-4, 1e-2}
>   * The pre-attention rank-1 projector with $\eta_{sic}$ = 0.5 and $\epsilon$ = 1e-4
>
> App. E.5 (Table 7) contains this study, but we agree that the main text should state the actual selection procedure and defaults more clearly.
>
> ### **Regarding Q2 (why the gains are tied to the proposed operators)**
>
> The appendix includes cross-substitution experiments to test whether the gains come from the proposed operators rather than generic tuning:
>   * Adding SIC to MSGNN improves accuracy by +0.5 to +1.6 %
>   * Adding GET to MagNet yields gains up to +3.2 %
>   * Replacing DirGNN/TFE scalar reweighting with our sign-aware gate gives consistent gains as well.
>
> Conversely, swapping baseline components back into GESC degrades performance. We will move this evidence forward in the paper because it directly supports the operator-level novelty claim.
>
> ### **Regarding Q3 (higher-rank bundles)**
>
> On the higher-rank bundle question, we strongly agree that this is an important direction. In this paper, we study the Abelian $U(1)$ case since it is the minimal non-trivial setting in which the cancellation idea can be isolated and implemented cleanly. But the principle is not inherently restricted to Abelian $U(1)$. A natural extension would replace scalar transport by $U_{ji} \in U(k)$ and replace the rank-1 projector by a projector onto a receiver subspace. Therefore, we view the current paper as establishing the cancellation principle first in the simplest gauge setting, not as claiming that the Abelian case is the end of the story.
>   * We agree that higher-rank fibers may offer additional expressive power, and we will state this more clearly as an important future direction.
>
> ### **Regarding Limitations**
> We thank the reviewer for this profound insight, which closely aligns with our response to Q3. Following your valuable feedback, we will explicitly incorporate this into a Limitations and Future Work discussion. We will clearly state that while the current Abelian formulation provides a computationally feasible foundation, its restriction to rank-1 fibers is a limitation, and extending this interference cancellation mechanism to higher-rank non-Abelian connections remains an important open challenge for the community.
>
> ### **Concluding Remarks**
> Thank you again for improving the quality of the paper.

---

> > ### Author Rebuttal · Reviewer_ups2 · 2026-04-04
> >
> > We highly appreciate the authors' excellent rebuttal answers and honesty.  Overall everything looks good.
> >
> > Here are just a couple of minor questions:
> >
> > 1)In your rebuttal of comparing to sheaf methods, did you guys choose the sheaf of only rank one? If this is the case, then it is still difficult to deflect the potential value of sheaf methods, because the higher rank sheaves still have other strengths.
> >
> > However, given your results that your methods outperform many scenarios of sheaf methods (even though with rank only being one), it still shows a potential of this projector-cancellation approach, so its generalization to higher rank bundles can be a big hope to overcome the "big computation head" issue in sheaf methods.
> >
> > 2)In your choice of the tuning parameters eta and epsilon, you did answer how you chose them in your experiments, but is there a principle on how to choose them in general situations?

---

> > > ### Author Response · Authors · 2026-04-04
> > >
> > > Dear Reviewer ups2,
> > >
> > > Thank you again for the thoughtful follow-up and for the encouraging perspective on the higher-rank direction. We agree with the meaning of both points.
> > >
> > > **1) On the sheaf comparison (higher-rank issue)**
> > >
> > > We fully recognize the importance of this direction, and we are familiar with the sheaf-based line of work, including its higher-rank formulations and their advantages. Our added sheaf comparison was meant as a representative empirical reference, not the value of the broader higher-rank sheaf family. In particular, we would like to state that the projector-cancellation principle itself is already useful in the minimal Abelian / rank-1 setting. We fully agree that higher-rank sheaves can have additional strengths due to richer local restriction maps and larger fiber dimension, and our current results should not be interpreted as ruling them out.
> > >
> > > More importantly, we view the two directions as complementary rather than competing. A natural extension of our framework is to replace scalar $U(1)$ transport with $U(k)$-valued transport and replace the current rank-1 projector by a projector onto a receiver subspace. In that sense, the current paper should be viewed as establishing the cancellation principle in the simplest clean setting, not as claiming that the full higher-rank (non-Abelian) has already been covered. We will revise the wording to make this limitation and future direction explicit, and to report the comparison setting more clearly. Again, thanks for pointing this out.
> > >
> > >
> > > **2) On a general principle for choosing $\eta_{sic}$ and $\epsilon$**
> > >
> > > Yes, there is a principle beyond simply saying we tuned them on validation. The key point is that SIC scales the receiver-parallel component by Prop. 5.1. So $\eta_{sic}$ directly controls the desired attenuation strength of the self-parallel part.  This is also why we keep $\eta_{sic}\in[0,1]$. In that range, SIC remains non-expansive and preserves the stability / Lipschitz guarantees in Sec. 5. Practically, a moderate value is the safest default. If oversmoothing / self-reinforcement is stronger (e.g., deeper propagation or more heterophilic neighborhoods), one should move $\eta_{sic}$ upward. If a useful self-aligned signal should be retained more strongly, one should move it downward. Our ablation suggests that $\eta_{sic}=0.5$ is a good default because it removes a substantial fraction of the parallel component without over-canceling it.
> > >
> > > The role of $\epsilon$ is different. It is mainly a numerical stability scale for the projector denominator. In general, $\epsilon$ should be small relative to the typical node-state energy so that SIC behaves like a projection on ordinary nodes (but not so small that very low norms make the projector unstable). Therefore, a practical rule is to choose $\epsilon$ as a small fraction of a robust statistic of $\|h_i\|_2^2$ (e.g., epoch median), or simply use a small fixed value when hidden-state norms are reasonably normalized. This is why $\epsilon=10^{-4}$ worked well in our experiments, which can avoid instability, but not change the intended projection behavior.
> > >
> > > ### **Concluding Remarks**
> > >
> > > We appreciate these insightful follow-up questions, as they help clarify both the scope (rank-1 vs. higher-rank) and the practical usage of SIC. We will revise the manuscript to better reflect these points, including clearer positioning with respect to higher-rank sheaf models and more explicit guidance on hyperparameter selection.
> > >
> > > Thank you again for the constructive feedback.

---

### Official Review · Reviewer_pRZD · 2026-03-12

**Soundness:** 3
**Presentation:** 2
**Significance:** 2
**Originality:** 2
**Overall Recommendation:** 5
**Confidence:** 3

**Summary:**

The paper introduces a projection-based self-interference cancellation mechanism by modeling interferences arising from low-frequency signals. The MPNNs addressing self-interference for graphs in a heterophilic regime struggle with representing the phase structure of the self-interactions due to the scalar re-weighting schemes adopted in the aggregation functions. The paper addresses the issues by directly cancelling the interactions. Specifically, the authors realize the cancellation mechanism with a rank-1 projection operator onto a given latent vector. This mechanism is reinforced through the sign-aware gating as well as both of the residual gating and attention mechanisms. The authors also provide various theoretical results to support the effectiveness of the mechanisms in the interference cancellation. The proposed method is evaluated across different standard scenarios widely used in the graph learning community and shows superior performance in the respective experiments.

**Compliance With Llm Reviewing Policy:**

Affirmed.

**Final Justification:**

Given the authors' update plans for the paper as well as the response to the other reviewers' comments, I am convinced that the paper's message became much clearer. As far as I am concerned, there is also no critical technical flaw in the paper, and its contribution is solid.

**Key Questions For Authors:**

1) What is the intuition and/or intention behind enforcing symmetry preservation on the model? What exactly does the phase $\{\psi_{i}\}$ represent in these applications, and what does the gauge transformation represent in respective applications?

2) Is the Lipschitz constant used for discussing oversmoothing related with certain implicit notions, for example, the Dirichlet energy-based oversmoothing metric defined as the symmetric normalized graph Laplacian?

3) What is the phase scale $\alpha$ in the equivariance test?

**Limitations:**

Yes.

**Strengths And Weaknesses:**

**Strength:**
- Solid performance gain across the standard graph-learning datasets.
- Crisply detailed proofs of theoretical results.


**Weakness:**

- Indication of the theoretical results is unclear. Every statement includes a sort of explanation and/or interpretation about theoretical results, but they are mostly just paraphrases of the results. I do not see them as informative as the authors' intention.

- The statement about the oversmoothing experiment does not sound convincing. Oversquashing can also happen in this context. The number of layers generally does not work as a decisive indicator of oversmoothing.  Furthermore, the reason behind using the Lipschitz constant as a metric for oversmoothing is unclear. I do not see a clear connection between the Lipschitz constant and oversmoothing.

- I am not sure whether the experiment conducted in 6.3 evaluates the performance of the model under edge perturbations. To me, this experiment looks like just performing a hyperparameter search.

- Descriptions on the theoretical results are generally loose. Some parts of the proofs of the theoretical results also sound repetitive and unnecessarily detailed. Besides, terminology is also a tad loosely introduced. Various new terminologies are used throughout the paper, but it was hard to tell which terminology specifies which mathematical equation. For example, I am still not very sure which part corresponds to the SIC mechanism. These worsen the readability. The followings are a couple of points that caught my attention:
  - $||$ in Proposition 5.1, which judging from the statements means linear independent vectors, is not a very common notation.
  - The equation (104) is identical to (93).
  - Please define $L_{GECS}$ and $L_{GECS}^{ || }$. I speculate $L_{GECS}$ is the Lipschitz constant of a GECS layer $F$, but I couldn't find the definition of $F$. Neither for $F_{||}$.
  - The equation (10), the upperscript of the left hand side should be (t) or (t+1).
  - The first equation (11) is essentially a repetition of (4).
  - Line 1120: $\lVert \tilde{h}^{(m)} _{j \rightarrow i} \rVert _{2}$ should be replaced with $\lVert r _{j \rightarrow i} \rVert _{2}$.

---

> ### Author Rebuttal · Authors · 2026-03-26
>
> Dear Reviewer pRZD,
>
> Thank you for the review. We agree that the main issues are: (i) theorem interpretations are too close to the formal statements, (ii) our over-smoothing language is broader than what the theory/experiments support, (iii) Sec. 6.3 is a sensitivity study, not an edge-perturbation test, and (iv) several notation/presentation issues reduce readability.
>
> ### **Regarding weakness 1 (unclear theoretical results)**
>
> Thank you. We will rewrite the theorem summaries in operational terms. Specifically:
> * Proposition 5.1 / Lemma 5.2: SIC is selective, not global damping; it attenuates only the component in span{$h_i^{(t)}$} and leaves the orthogonal component unchanged.
> * Proposition 5.4: a per-head magnitude-control result for gated aggregation.
> * Theorem 5.5: node-wise phase reparameterization (gauge) does not change the scalar comparisons used by transport, SIC, gating, or attention.
> * Theorem 5.8: a layerwise stability bound, with tighter control along self-aligned directions, rather than a full theorem of over-smoothing.
>
> ### **Regarding weakness 2 (over-smoothing experiment)**
>
> We agree. The current depth experiment does not disentangle over-smoothing from oversquashing, and Theorem 5.8 is not a Dirichlet-energy/Laplacian over-smoothing metric. Therefore, we will narrow the claim as follows:
> * We will replace broad statements such as primary driver of oversmoothing, mitigates over-smoothing, and similar wording in the Abstract/Intro/Conclusion with the narrower statement that SIC reduces self-parallel amplification and improves depth stability relative to additive baselines.
> * We will relabel Remark 5.3 as heuristic spectral intuition, and describe Fig. 4 only as a depth-stability illustration.
>
> ### **Regarding weakness 3 (Sec. 6.3 robustness)**
>
> We apologize for the confusion. Sec. 6.3 is a sensitivity analysis over $\eta_{\mathrm{sic}}$ and $\lambda_{\mathrm{JS}}$, not an edge-perturbation robustness test.
> * We will rename Sec. 6.3/Q3 accordingly and remove the current edge-perturbation wording.
> * App. E.4 will be presented only as a gauge-consistency test under node-wise phase perturbations, not as general structural robustness.
>
> ### **Regarding weakness 4 (theoretical descriptions / notation)**
>
> We agree and will tighten terminology and notation. Concretely:
> * We will add a short module-to-equation map: GET (Eq. 10), SIC projector (Eq. 11), SIC cancellation (Eq. 12), sign-aware gate (Eqs. 15–17), residual gate (Eq. 18), and post-gate message (Eq. 19).
> * In Proposition 5.1, we will replace the current decomposition wording with explicit span{$h_i^{(t)}$} / orthogonal-complement notation.
> * We will define $F$ explicitly as one GESC layer. There is no separate $F_{\parallel}$, and $L_{\mathrm{GESC}}^{\parallel}$ refers to the directional Lipschitz constant of $F$ restricted to self-parallel perturbations.
> * We will remove duplicated equations, fix the Eq. 10 layer superscript, mark Eq. 11 as a recalled projector definition, fix the line-1120 typo, and shorten repetitive proof text in Appendix C.
>
> ### **Regarding Q1**
>
> Thanks for pointing this out. We do not enforce gauge equivariance as a symmetry prior for its own sake. Its role is to make the comparison between a receiver state and a transported neighbor message well-defined after complex transport. On these benchmarks, the learned phase $\theta_{ji}$ in $U_{ji}=e^{i\theta_{ji}}$ is best viewed as a latent relative orientation/phase parameter, not a directly observed physical quantity. A gauge transformation $\psi_i$ is a node-wise change of local complex reference frame. Equivariance ensures that the same latent state (expressed in different local phase conventions) yields the same scalar decisions for transport, SIC, gating, and attention.
>
> ### **Regarding Q2**
>
> Not in the strict sense. $L_{\mathrm{GESC}}$ is the global Lipschitz bound of one layer $F$, and $L_{\mathrm{GESC}}^{\parallel}$ is the corresponding directional bound along self-parallel perturbations. These are not defined through the symmetric normalized graph Laplacian and are therefore not equivalent to Dirichlet-energy-based over-smoothing metrics. Our theory supports a narrower statement: SIC tightens control along self-aligned directions, but does not by itself fully characterize over-smoothing.
>
> ### **Regarding Q3**
>
> $\alpha$ is the perturbation amplitude in App. E.4. We sample node-wise phases $\phi_i \sim U[0,2\pi]$, apply $h_i \leftarrow e^{i\alpha\phi_i}h_i$ and $U_{ji} \leftarrow e^{i\alpha(\phi_j-\phi_i)}U_{ji}$, and then measure prediction agreement, logit $\ell_2$ drift, and attention KL divergence. Thus, $\alpha=0$ means no perturbation and $\alpha=1$ means the full sampled random gauge transformation.
>
> ### **Concluding Remarks**
>
> We sincerely thank the reviewer again. In the revision, we will narrow the theoretical/empirical claims to what is directly supported, clarify the role of gauge equivariance, and fix the notation/presentation issues you identified.

---

> > ### Author Rebuttal · Reviewer_pRZD · 2026-04-03
> >
> > I thank the authors for the rebuttal. My concerns were fully addressed. Given the authors' update plans for the paper as well as the response to the other reviewers' comments, the paper's message became much clearer and convincing to me. As far as I am concerned, there is also no critical technical flaw in the paper, and its contribution is solid. Therefore, I will recommend the acceptance, and raised my score accordingly.

---

> > > ### Author Response · Authors · 2026-04-03
> > >
> > > Dear Reviewer pRZD,
> > >
> > > Thank you again for your detailed feedback and valuable suggestions for improvement.
> > >
> > > We also appreciate you pointing out issues that we should have identified earlier.
> > >
> > > Best regards,
> > >
> > > Authors

---

### Official Review · Reviewer_PeNt · 2026-03-25

**Soundness:** 3
**Presentation:** 3
**Significance:** 3
**Originality:** 3
**Overall Recommendation:** 4
**Confidence:** 3

**Summary:**

The authors focus on addressing the failures of GNNs when dealing with the heterophily, inconsistent phase and over-smoothing. The work argues that traditional additive aggregation can reinforce the low-frequency components shared between target node and self-parallel neighbor messages, leading to a cumulative bias and severe over-smoothing. Therefore, the authors propose GESC (Graph-Equivariant Graph Network with Self-Interference Cancellation), which transfers neighboring messages into the target node’s reference frame, and then utilize Tikhonov regularization and a rank-1 projection to remove self-parallel components. The GESC also introduces a sign-aware and residual gate to determine how the transported message and the residual signal are combined.
Overall, this work proposes a new solution to several longstanding graph problems, including over-smoothing, and performance degradation under heterophily. The methods are thoroughly validated through comprehensive experiments on both medium-scale and large=scale graph dataset, and achieve SOTA or top-3 on most of them.

**Compliance With Llm Reviewing Policy:**

Affirmed.

**Key Questions For Authors:**

No

**Limitations:**

Yes

**Strengths And Weaknesses:**

**Strengths**
- The description of methodology in the paper is clear. The logical chain from transported source to self-interference cancellation (SIC), sign-aware gating, and the gauge transport mechanism is complete. The functionalities and principles of these proposed modules are adequately explained and theoretically supported in the section 5. For example, section 5.1 explains why SIC works in a clear manner. Additionally, the stability given by the sign-aware gating design and motivation for introducing complex-valued transport are also elaborated in the subsequence sections. The overall story in this work is complete and sound.
- For presentation, the structure of the paper is clear. Necessary components including target questions, motivation, related work, preliminaries, method, theory and experiments are included and properly ordered. The figures and tables are clear and easy to follow.
- For significance, this work addresses several important problems, such as over-smoothing, heterophily-related failure. It introduces a compete methodology and justifies its effectiveness through multiple experiments. The strong results on heterophily datasets like Chameleon, Squirrel or Cornell or large graph datasets such as OGB demonstrate the effectiveness of the method.
- Additionally, the work combines complex-valued representation, gauge-equivariant transport, attention and anti-oversmoothing mechanisms into a new message-passing method in a logic and effective way, showing originality.

**Weaknesses**
- It is claimed that the lack of interference handling, is a primary driver of over-smoothing under gauge transport. Yet, the evidence provided in the paper is not sufficient. The SIC’s effectiveness is supported by ablation experiments and is highly likely related to its ability to mitigate over-smoothing. However, this still does not fully support the claim that interference is the main factor leading to over-smoothing. For example, SIC’s effectiveness may come from the improved feature diversity, better geometric representation or greater stability.
- The modifications of GNN proposed in this work are likely to introduce additional computation overhead. The paper provides theoretical analysis of this issue, yet it lacks sufficient experiments and corresponding results to demonstrate that these methods remain effective in practical settings.

---

> ### Author Rebuttal · Authors · 2026-03-26
>
> Dear Reviewer PeNt,
>
> Thank you for the careful and thoughtful review. We appreciate your positive assessment of the method, presentation, and empirical analysis.
>
> ### **Regarding weakness 1 (self-interference handling)**
>
> * We agree that our wording (interference leads to over-smoothing) is quite strong. The current evidence supports a narrower claim:
>   * Under gauge-consistent transport followed by additive aggregation, repeated buildup of the receiver-aligned component is a major contributor to self-reinforcement, and canceling that component improves depth stability.
>   * It does not establish that interference handling is the unique cause of oversmoothing in general. Therefore, we will redefine it as a "major contributor under additive gauge transport", and revise the abstract/introduction/theory discussion accordingly.
>
> * We also agree that SIC’s (Self-Interference Cancellation) gains could stem from improved diversity or stability. However, we claim that, in the present model, these effects arise from a direction-specific mechanism rather than from generic extra capacity.
>   * Proposition 5.1 shows that SIC contracts only the component parallel to the receiver state while leaving the orthogonal component unchanged
>   * Lemma 5.2 shows that the self-parallel energy does not increase
>   * Theorem 5.8 tightens the Lipschitz factor specifically along self-aligned directions.
>
> * In the ablation study, we have shown the following results. However, as you mentioned, we will revise the wording to attribute the effect to this specific self-aligned buildup mechanism.
>   * Removing SIC lowers accuracy on all four datasets (Table 2)
>   * In the SIC projector ablation (App. E.5 / Table 7), the best setting is a pre-attention rank-1 projector with moderate cancellation, while stronger cancellation, post-attention placement, or higher-rank projection performs worse
>   * In the cross-substitution study (App. E.6 / Table 8), adding SIC to MSGNN yields consistent gains (from +0.5% to +1.6%).
>
>
> ### **Regarding weakness 2 (practical view)**
>
> Thank you for raising this. We agree that the main paper does not handle this clearly enough. However, please note that we have provided two relevant pieces of evidence in the appendix.
>   * First, App. B.3 gives the layer complexity as $O(MEd^2)$ in the direct form. Thus, the method remains sparse and linear in the number of edges up to the standard feature-transform cost. The implementation uses sparse aggregation + vectorized SIC, and attention over edges.
>   * Second, we included large-graph results precisely to test practicality: App. E.1 reports Penn94, arXiv-year, and snap-patents, where GESC is best among the reported baselines and remains trainable on snap-patents, where H2GCN runs out of memory. Further, we report OGB-scale results on ogbn-arxiv, ogbn-proteins, and ogbn-mag (App. E.7).
>
> To be fully precise, we do not currently claim wall-clock superiority over simpler real-valued GNNs. Instead, we claim that the method remains practically trainable at large scale and that its added operators are sparse-linear rather than combinatorial. In the revision, we will (i) narrow the causal oversmoothing claim, and (ii) move the existing complexity and large-scale evidence into the main paper.
>
> ### **Concluding Remarks**
>
> Thank you again for identifying exactly where the presentation should be tightened.

---

### Decision · Program_Chairs · 2026-04-30

**Decision:**

Accept (regular)

**Comment:**

This paper proposes a gauge-based architecture to improve learning on heterophilic graphs. The authors argue that standard message passing in GNNs (with sum or mean aggregation) inherently fails to appropriately handle interference of heterophilic node representations during (node-wise) aggregation of messages. The paper addresses the issues by directly cancelling the node interactions. More explicitly, the authors propose a gauge connection together with a cancellation mechanism obtained via a rank-1 projection that suppresses self-parallel components. Furthermore, the authors introduce sign-aware gating to regulate contributions from neighbors with negative alignment. Overall, this work proposes a new solution to several longstanding graph problems, including over-smoothing and performance degradation under heterophily. The methods are thoroughly validated through comprehensive experiments on both medium-scale and large-scale graph datasets, and achieve SOTA or top-3 on most of them.

All reviewers agree that the paper makes a meaningful contribution to the field. The work is well-motivated, and the proposed approach is clearly presented and supported by a solid theoretical and empirical evaluation.
During the rebuttal phase, the authors thoroughly addressed all concerns raised by the reviewers. Two have updated their assessments and voted in favor of acceptance. In light of this consensus and the demonstrated quality of the work, I recommend acceptance.